# Bullet Trains: Parallelizing Training of Temporally Precise Spiking Neural Networks

**Todd Morrill** [1 2]  **Christian Pehle** [2]  **Anthony Zador** [2]

## Abstract

Continuous-time, event-native spiking neural networks (SNNs) operate strictly on spike events, treating spike timing and ordering as the representation rather than an artifact of time discretization. This viewpoint aligns with biological computation and with the native resolution of event sensors and neuromorphic processors, while enabling compute and memory that scale with the number of events. However, two challenges hinder practical, end-to-end trainable event-based SNN systems: 1) exact charge–fire–reset dynamics impose inherently sequential processing of input spikes, and 2) precise spike times must be solved without time bins. We address both. First, we use parallel associative scans to consume multiple input spikes at once, yielding up to 44x speedups over sequential simulation while retaining exact hard-reset dynamics. Second, we implement differentiable spike time solvers that compute spike times to machine-precision without discrete-time approximations or restrictive analytic assumptions. We demonstrate the viability of training SNNs using our solutions on four event-based datasets on GPUs.

## 1. Introduction

A commonly cited advantage of spiking neural networks (SNNs) is their ability to operate in an event-based manner, processing information only when spikes occur (Merolla et al., 2014; Davies et al., 2018). On appropriate neuromorphic hardware, this implies energy efficiency, as computation only occurs when spikes are present (Roy et al., 2019; Yao et al., 2024). At the same time, much SNN

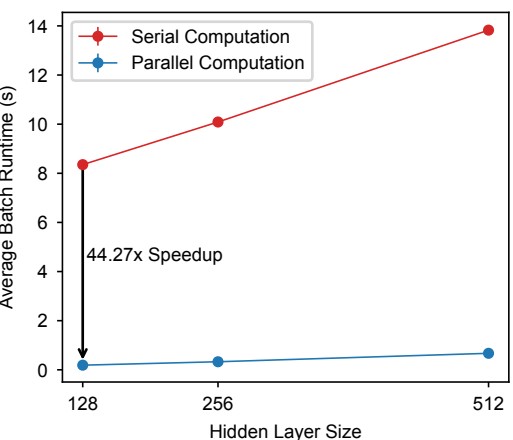

*Figure 1.* Our parallel method achieves up to 44x speedups over serial processing of spike events on several event-based datasets while retaining exact hard-reset dynamics. The plot shows results for the Spiking Heidelberg Digits (SHD) dataset.

research still happens on readily available hardware such as graphical processing units (GPUs) so accelerating this workload (e.g., via parallelization) and mirroring properties of biology and event-based neuromorphic hardware (e.g., precise spike times) (Carr & Konishi, 1990; Thorpe & Gautrais, 1998; Amir et al., 2017; Richter et al., 2024) is an important research direction. In this work, we present a strictly event-based SNN system that processes multiple spike events in parallel, achieving significant speedups over sequential processing. Second, we present differentiable spike time solvers that enable machine-precision spike timing. These two contributions allow us to take a step toward event-native spiking computation, where preserving continuous spike times is the objective—both for biological fidelity and for compatibility with neuromorphic sensing and hardware—and where efficiency follows from operating directly on events.

SNNs are challenging to parallelize along the time dimension due to the sequential nature of "charge–fire–reset" dynamics. After consuming each and every input spike, a neuron must determine whether it will produce a spike itself before the next input spike arrives. Performed naively, SNNs must process one spike at a time, leading to signifi-

[1]Department of Computer Science, Columbia University, New York, NY, USA [2]Department of Neuroscience, Cold Spring Harbor Laboratory, Cold Spring Harbor, NY, USA. Correspondence to: Todd Morrill <todd@cs.columbia.edu>.

*Proceedings of the 43rd International Conference on Machine Learning*, Seoul, South Korea. PMLR 306, 2026. Copyright 2026 by the author(s).

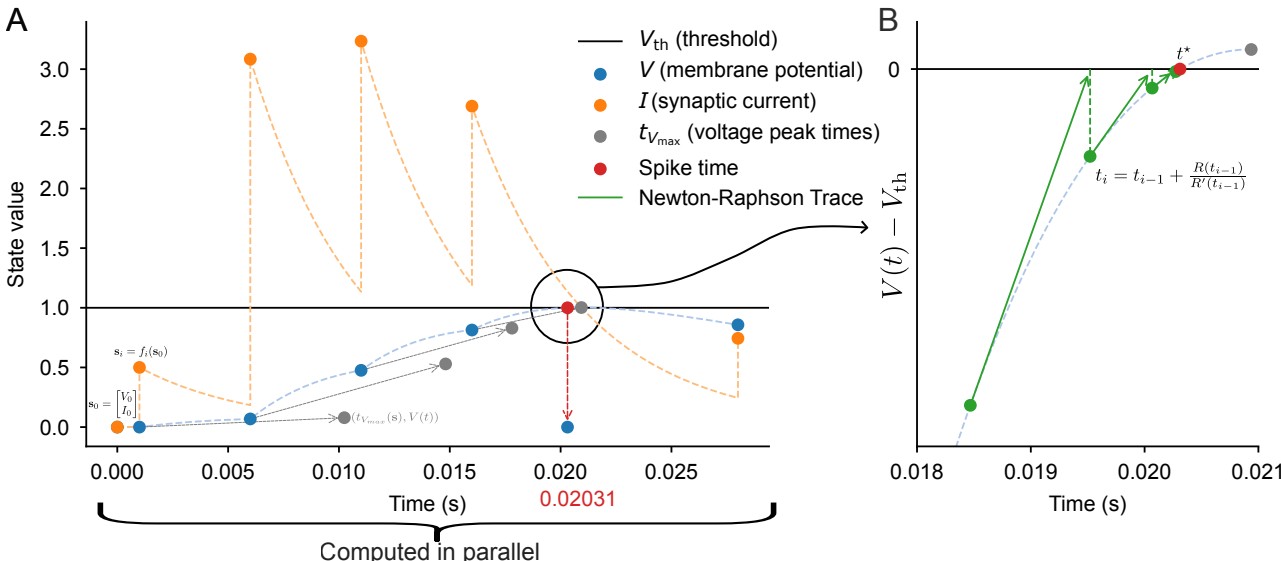

*Figure 2.* Our contributions visualized—state trajectory of a LIF neuron consuming input spikes in parallel and producing an output spike with our Newton-Raphson solver. **A** shows the neuron jumping from the initial state $\mathbf{s}_0$ at time $0$ (composed of membrane potential and synaptic current) directly to all future states $\mathbf{s}_1 = f_1(\mathbf{s}_0), \mathbf{s}_2 = f_2(\mathbf{s}_0), \ldots$ in parallel using associative scans. We use lightweight analytical checks to determine if an output spike will occur in the interval between any consecutive pair of input spikes by computing—in parallel—$t_{V_{\max}}(\mathbf{s})$, the time of maximum voltage starting at the interval's left endpoint, and $V(t_{V_{\max}})$ the voltage at that time. The voltage state is hard reset at the spike time. **B** shows our iterative Newton-Raphson solver finding the output spike time $t^\star$, computed to machine-precision.

cant computational bottlenecks on modern parallel hardware such as GPUs (Shrestha & Orchard, 2018; Engelken, 2023; Fang et al., 2023; Li et al., 2024; Zhong et al., 2024; Feng et al., 2025). In the past few years, many methods have been proposed to parallelize spiking dynamics, but they all modify the neuron model in some way, such as removing resets entirely, which can reduce the non-linear expressivity of the neuron and/or deviate from the way biological neurons operate.

In this work, we propose a method for parallel training of exact hard-reset spiking dynamics using parallel associative scans. The key idea is to process input spike trains in chunks, where each chunk is processed in parallel using associative scans. Crucially, this is spike-aware chunking: lightweight analytical checks quickly determine whether a neuron spikes within a chunk and, if so, locate the first such spike, allowing the simulation to jump directly to that time. The next chunk resumes processing input spikes immediately after the output spike. In practice, even if a portion of work is discarded, the overall speedup relative to sequential processing is significant because of the parallelism afforded by modern hardware. The result is a parallel system that achieves up to 44x speedups over sequential processing on several event-based datasets while retaining exact hard-reset dynamics.

A second challenge that arises in strictly event-based SNNs is determining spike times precisely without the use of discrete-time bins. Nearly all SNN implementations to date operate on discrete-time grids (Bauer et al., 2023; Hammouamri et al., 2024; Nowotny et al., 2025), which means that neurons are performing computation at every time step, regardless of whether spikes are present. The result of this modeling choice is computation—and often memory usage—that scales with the number of time steps rather than the number of spikes, severely limiting the number of time steps that can be simulated. Temporal resolution is limited by the time-step size and spike ordering within a time bin cannot be determined (Bauer et al., 2023) (see Figure 3). In contrast, several systems have proposed analytical solutions for continuous spike times, but these methods often impose restrictions on the neuron model, such as the relationship between synaptic and membrane time constants in leaky-integrate-and-fire (LIF) neurons (Mostafa, 2018; Göltz et al., 2021; Engelken, 2023).

Here, we apply several methods for differentiable spike time solvers—namely Newton-Raphson and Bisection numerical root solvers—which resolve spike times to machine-precision without the need for discrete-time approximations. This removes a constraint on SNN systems (e.g., to support heterogeneous time constants (Perez-Nieves et al., 2021)), facilitates the exploration of neuro-inspired theories of time-based computation (Yang & Zador, 2012; Xie et al., 2024), matches the temporal resolution of event-based sensors and neuromorphic hardware (Müller et al., 2024), and enables compute and memory to scale in proportion to the number

of spikes.

## 2. Related Work

**Parallelizing Spiking Dynamics.** Biological neurons go through a "charge–fire–reset" cycle and in the "reset" stage, the neuron's membrane potential is set to a baseline value (Hodgkin & Huxley, 1952). It has been argued that this hard reset is essential for implementing precise feature detection (Berry & Meister, 1998). The sequential dependency of the "charge–fire–reset" cycle in simulated Leaky Integrate-and-Fire (LIF) neurons presents a fundamental barrier to parallel training on modern hardware. While sub-threshold dynamics are linear and can be parallelized using associative scans—similar to state-space models (Schöne et al., 2025)—the non-linear spike generation and hard reset introduce a sequential dependency that prevents complete parallelization. Early approaches, such as the Parallel Spiking Neuron (PSN), circumvented this "charge–fire–reset" cycle by removing the reset mechanism entirely, effectively reducing the neuron to a linear filter solvable via convolutions (Fang et al., 2023). While efficient, this sacrifices the non-linear regulation (i.e., "hard reset") of the membrane potential essential for expressivity and biological fidelity (Bauer et al., 2023). Subsequent methods like the Parallel Spiking Unit (Li et al., 2024) and SPikE-SSM (Zhong et al., 2024) decoupled the reset from integration, but rely on "soft reset" mechanisms (linear subtraction) rather than "hard reset" (reset-to-zero). Most recently, Fixed-point Parallel Training (FPT) attempted to model hard resets via iterative scans, yet to guarantee convergence of the fixed-point iteration, FPT must relax the discontinuous spike generation into a continuous sigmoid surrogate during the forward pass (Feng et al., 2025). Consequently, existing parallel methods either fundamentally alter the neuron model (removing resets), assume linearity (soft resets), or solve a relaxed approximation (smoothed dynamics). To the best of our knowledge, no prior work has achieved parallel acceleration of exact, hard-reset dynamics without approximation.

**Precise Spike Time Solvers.** SNN implementations must make two core design choices: 1) whether to use surrogate gradients or exact gradients, and 2) whether to use a discrete-time grid or operate in continuous time. If exact gradients are coupled with continuous time, a third decision must be made: whether to use analytical spike time solutions or numerical spike time solvers. Surrogate gradient (SG) methods (Shrestha & Orchard, 2018; Neftci et al., 2019) are inextricably linked to discrete time steps. We therefore focus our discussion on exact gradient methods to retain machine-precision spike times. Many exact gradient methods to date operate on discrete-time grids, where spike times are approximated to the nearest time step (Bauer et al., 2023; Nowotny et al., 2025; Mészáros et al., 2025). Several

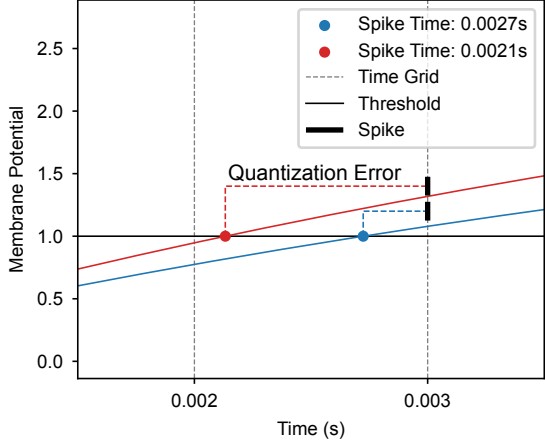

*Figure 3.* Discrete-time binning causes quantization errors and loses spike ordering. Here, two spikes occurring at different times (0.0021s and 0.0027s) both round to 0.003s, making them indistinguishable to downstream neurons.

methods have used analytical solutions for continuous spike times, but these methods often impose restrictions on the neuron model (Bohté et al., 2000; Mostafa, 2018; Göltz et al., 2021; Engelken, 2023). Newton-Raphson and Bisection methods have been proposed for finding spike times in neural simulations (D'Haene et al., 2009) but have not been demonstrated for training SNNs with gradient descent on large-scale tasks on GPUs.[1] In this work, we demonstrate the effectiveness of training SNNs using numerical spike time solvers based on the Newton-Raphson and Bisection methods, which can be applied to a wide variety of neuron models—without restrictive assumptions—while working with exact gradients in continuous-time.

## 3. Event-Based Spiking Neural Networks

Here we describe our proposed methods for building event-based spiking neural networks that 1) process multiple input spike events in parallel while maintaining temporal causality with exact hard-reset dynamics, and 2) determine spike times precisely using differentiable numerical spike time solvers.

### 3.1. LIF Neurons

Our SNNs are built from current-based LIF neurons (Rotter & Diesmann, 1999; Gerstner & Kistler, 2002), one of the most popular neurons in the SNN community (Wunderlich & Pehle, 2021; Göltz et al., 2021; Nowotny et al., 2025; Hammouamri et al., 2024; Mészáros et al., 2025), with the following sub-threshold dynamics—the dynamics between

---

[1]Wunderlich & Pehle (2021) used the TOMS 748 (Alefeld et al., 1995) root solving algorithm on a CPU.

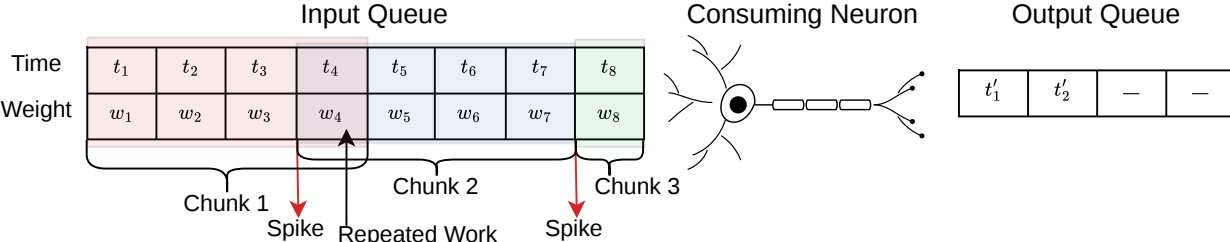

*Figure 4.* An example input queue feeding a neuron and producing an output queue. The input chunk size is 4 spikes, which means 4 input spikes are consumed in parallel. After consuming 3 input spikes in chunk 1, the neuron produces an output spike. The work done to consume the 4th input spike in chunk 1 is discarded, and the next chunk resumes processing input spikes immediately after the output spike time.

spikes—on the synaptic input current $I(t)$ and membrane potential $V(t)$.

$$\tau_m \frac{dV}{dt} = -V(t) + I(t) \tag{1}$$

$$\tau_s \frac{dI}{dt} = -I(t) \tag{2}$$

where $\tau_m$ and $\tau_s$ are the membrane and synaptic time constants, respectively. Closed-form solutions for these differential equations are provided in Appendix Section A.

A LIF neuron emits a spike at time $t$ when the membrane potential crosses a threshold $V_{\text{th}}$ (i.e., $V(t) \geq V_{\text{th}}$), as depicted in Figure 2A and then resets to a reset potential $V_{\text{reset}}$, which is set to 0 in this work. Neurons indexed by $i$ that are downstream of the spiking neuron $j$ have their synaptic input currents incremented by the synaptic weights $w_{ij}$, which are learned during training. We also introduce learnable synaptic delays $d_{ij}$, which simulate the time a spike would take to be transmitted from one neuron to the next. The synaptic input current is then

$$I(t) \leftarrow I(t) + \sum_k w_{ij}\delta(t - (t_k + d_{ij})) \tag{3}$$

where $t_k$ is the time of the $k^{\text{th}}$ spike from neuron $j$ and $\delta(t)$ is the Dirac delta function.

### 3.2. Event-Driven Implementation with Exact Gradients

We implement our SNN in `JAX` in an event-based manner. The core idea of an event-based SNN is that if for each neuron in the system you know which of two events will happen first, namely whether a neuron should first consume an incoming spike or first emit a spike itself, then you can directly advance the simulation to the appropriate event. This highlights the distinction between our event-driven approach and discrete-time stepped approaches, which simulate the entire network's state (i.e., membrane potential and synaptic input current) at every discrete point in time. We train the model's parameters via gradient descent using

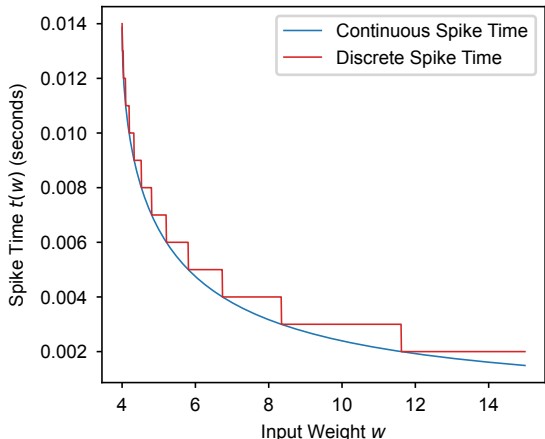

*Figure 5.* Spike time as a function of synaptic weight for different time constant configurations. Discrete-time approximation with $\Delta t = 1$ms diverges from the ground truth spike time at nearly all weight values, while our method closely tracks the ground truth spike time.

exact gradients. The key idea for being able to use exact gradient descent is that spike *times* are differentiable functions of the SNN's synaptic weights and delay parameters (Mostafa, 2018; Wunderlich & Pehle, 2021). Intuitively, if a weight onto a postsynaptic neuron is increased, it has the effect of charging the neuron up faster, and in turn accelerating the spiking time of the postsynaptic neuron (see Figure 5). The event-based exact gradient approach is in contrast to surrogate gradient methods, which use a discrete-time stepped approach and lose the ability to disambiguate spikes within the step window (see Figure 3) and therefore machine-precision spike times.

#### 3.2.1. PARALLEL PROCESSING OF SPIKE EVENTS

**Deriving the associative scan.** Processing incoming spikes to a LIF neuron one-by-one becomes a computational bottleneck because it scales linearly with the number of input spikes as $O(N)$ where $N$ is the number of spikes to be

processed. State-space models similarly face this problem due to their recurrent nature. One solution to this problem is to parallelize computation along the time-dimension through the use of an associative scan. The key observation is to note that sub-threshold transitions of the LIF neuron state—its voltage and input current—can be expressed as affine maps, which admit an associative operation on these maps. These affine maps allow one to compute a transition from an initial voltage and input current state, $\mathbf{s}_0 = [V_0, I_0]^\top$, to all neuron states in the future at each input spike time, $\mathbf{s}_1 = [V_1, I_1]^\top, \ldots, \mathbf{s}_N = [V_N, I_N]^\top$. Importantly, once these affine maps are computed, we can consume all input spikes in parallel using an associative scan and directly advance the neuron to any intermediate state (i.e., any state $\mathbf{s}_i$ for $i \in [1, N]$) without having to sequentially process all previous spikes. To understand how we derive the associative operator, it is instructive to see how the neuron's state transitions from the first state $\mathbf{s}_0$ to the second state $\mathbf{s}_1$ after consuming the first input spike. Let $C_m = \exp(-t/\tau_m)$ and $C_s = \exp(-t/\tau_s)$. The affine map $f_1 : \mathbf{s}_0 \rightarrow \mathbf{s}_1$ to transition the neuron from state $i = 0$ to state $i = 1$, for example, is given by

$$
\underbrace{\begin{bmatrix} V_1 \\ I_1 \end{bmatrix}}_{\mathbf{s}_1} = \underbrace{\begin{bmatrix} C_m & \frac{\tau_s}{\tau_m - \tau_s}[C_m - C_s] \\ 0 & C_s \end{bmatrix}}_{M_1} \underbrace{\begin{bmatrix} V_0 \\ I_0 \end{bmatrix}}_{\mathbf{s}_0} + \underbrace{\begin{bmatrix} 0 \\ w \end{bmatrix}}_{\mathbf{b}_1} \quad (4)
$$

where we have denoted $M_1$ as the matrix that advances the neuron's state, and $\mathbf{b}_1$ as the vector that injects a new spike into the neuron with weight $w$. Equation 4 follows from closed-form equations for our system (see Appendix Equations 23 and 24) and Equation 3. Note that $M_1$ is defined by the time between the starting time, $t_0$, and the spike time $t_1$ (i.e., $t = t_1 - t_0$).

**Implementing the associative scan.** Examining the composition of two consecutive affine maps (i.e., consuming 2 consecutive input spikes) shows that composing such maps yields another affine map, which leads us to an associative operator commonly called *Combine* in parallel scan algorithms (lemma and proof in Appendix Section B). $f_2 : \mathbf{s}_1 \rightarrow \mathbf{s}_2$ is given as

$$
\mathbf{s}_2 = f_2(\mathbf{s}_1) = M_2 \mathbf{s}_1 + \mathbf{b}_2 \quad (5)
$$

$$
= M_2(M_1 \mathbf{s}_0 + \mathbf{b}_1) + \mathbf{b}_2 \quad (6)
$$

$$
= \underbrace{M_2 M_1}_{M_2'} \mathbf{s}_0 + \underbrace{M_2 \mathbf{b}_1 + \mathbf{b}_2}_{\mathbf{b}_2'} \quad (7)
$$

where now we can define an affine map to map directly from $\mathbf{s}_0$ to $\mathbf{s}_2$ as

$$
\mathbf{s}_2 = f_2'(\mathbf{s}_0) = M_2' \mathbf{s}_0 + \mathbf{b}_2' \quad (8)
$$

From this we propose a form for the associative *Combine* operator as

$$
Combine((M_2, \mathbf{b}_2), (M_1, \mathbf{b}_1)) = (M_2 M_1, M_2 \mathbf{b}_1 + \mathbf{b}_2).
$$

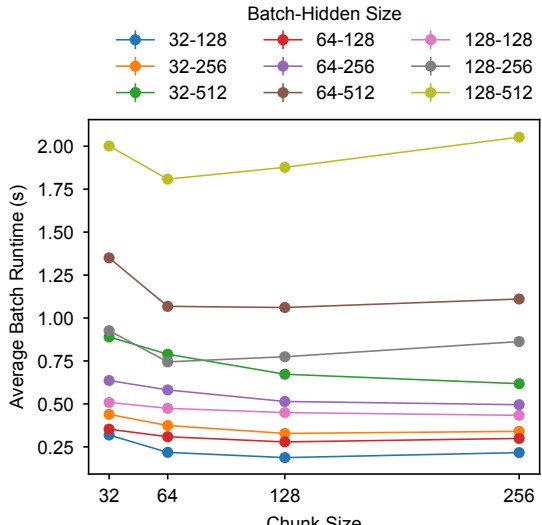

*Figure 6.* Training time per batch as we vary chunk size on the SHD dataset with hidden dimensions ranging from 128 to 512 and batch size ranging from 32 to 128. Averages are taken over 100 batches.

The important point is that *Combine* allows us to make use of the `jax.lax.associative_scan` function, which computes a sequence of $K$ affine maps, $f_1, f_2, \ldots, f_K$, with parallel depth $O(\log K)$—exponentially faster than the $O(K)$ serial approach because operations can be combined in a tree-like fashion (Blelloch, 1990). Once we have these affine maps, we can produce the sequence $(\mathbf{s}_0, \mathbf{s}_1, \mathbf{s}_2, \ldots, \mathbf{s}_K)$ in parallel. Since a neuron can produce a spike at any point after consuming a spike (see Figure 2) and we would like to avoid discarding excessive amounts of work, we consume input spikes in chunks, where each chunk contains $K$ input spikes that are processed in parallel using the associative scan approach, which implies a parallel compute graph depth of $O(C \log K)$ where $C$ is the number of chunks.

**Speculative execution.** We speculatively execute the associative scan for a chunk of input spikes (e.g., 128 spikes), which means we process the chunk without knowing whether an output spike will occur within the chunk. Between each pair of input spikes in a chunk, we determine if an output spike will occur in that interval (see Section 3.2.2). If the neuron spikes, we discard any work done within the chunk after the output spike time and revisit those input spikes on the next iteration (see Figure 4). Discarding work refers only to intermediate computations *after* the first within-chunk output spike; the event order and hard resets for processed spikes are unchanged.

**Fixed compute budget.** For efficient GPU just-in-time (JIT) compilation, we pre-specify a fixed compute budget—the number of chunks $C$ and a per-neuron output-spike cap $S_{\max}$ (Appendix Section E.2)—so all processed spikes follow exact hard-reset semantics, though if the cap activates, some input spikes may remain unprocessed within the compiled budget. In practice, we consume $\approx 100\%$ of SSC and SHD input spikes in layer 1 ($S_{\max}^{(1)} = 42$) and layer 2 ($S_{\max}^{(2)} = 28$) (Appendix Figures 11-13). Spike-count regularizers (Appendix Section E.1) keep firing sparse, maintaining high consumption without sacrificing accuracy.

### 3.2.2. DIFFERENTIABLE SPIKE TIME SOLVERS

Employing root solvers to find precise spike times introduces several sub-problems: 1) defining the function whose root corresponds to the spike time, 2) finding intervals that contain spikes to guarantee 100% exactness of the simulation, 3) choosing a root solver, and 4) deriving gradients through the root solver. We address each of these in turn.

**Defining the key relation for spike times.** A core capability of any SNN system is to detect the times at which a neuron produces spikes. This involves solving for $t^\star$, the time at which the membrane potential $V(t)$ crosses the threshold $V_{\text{th}}$. We use root solvers, which keep our system event-based and temporally precise. A spike occurs at the time at which the following relation $R$ is satisfied

$$R(V_0, I_0, V_{\text{th}}, t) = V(V_0, I_0, t) - V_{\text{th}} = 0 \qquad (9)$$

namely when the membrane potential $V$ is equal to the threshold, $V_{\text{th}}$, where $V(V_0, I_0, t)$ is the membrane potential at time $t$ given initial conditions $V_0$ and $I_0$ (see Equation 23). For brevity, we denote the neuron's initial state and threshold as $\mathbf{p} = (V_0, I_0, V_{\text{th}})$, allowing us to write the spike condition compactly as $R(\mathbf{p}, t(\mathbf{p})) = 0$, where $t(\mathbf{p})$ is the spike time as a function of these parameters.

**Finding intervals with spikes.** We can determine analytically *if* a spike will occur in the interval between two consecutive input spikes, which is essential for guaranteeing the exactness of our simulation. We do this by evaluating $V(t_{\text{now}})$ and $V(t_{\text{next}})$, the voltage at the start and end of the interval, respectively, as well as at the time of maximum voltage within the interval, $V(t_{V_{\max}})$. Note that $t_{\text{now}}$ is the left endpoint of the interval and $t_{\text{next}}$ is the right endpoint (a relative, not absolute, time). The parallel associative scan method provides us the needed neuron state values, $V(t_{\text{now}})$ and $V(t_{\text{next}})$. We can also evaluate the membrane potential at $t_{V_{\max}}$ to get $V(t_{V_{\max}})$ (see Appendix Section D.1 for a derivation). Crucially, we can now apply the following logic

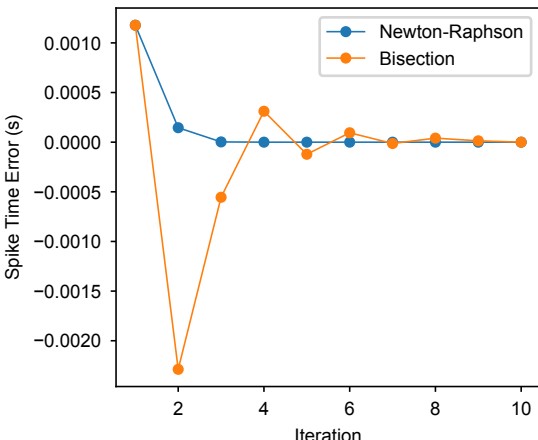

*Figure 7.* Convergence of the Newton-Raphson and Bisection solvers towards the ground truth spike time.

to determine if a spike occurs in the interval $[t_{\text{now}}, t_{\text{next}}]$:

If $V(t_{\text{next}}) \geq V_{\text{th}} \implies t^\star$ from root solver

If $V(t_{V_{\max}}) \geq V_{\text{th}}, t_{V_{\max}} \in [t_{\text{now}}, t_{\text{next}}] \implies t^\star$ from root solver

Else $\implies$ no spike in $[t_{\text{now}}, t_{\text{next}}]$.

**Root solvers.** There are many choices of root solvers for recovering the output spike time $t^\star$. Two effective options are Newton-Raphson, which iteratively refines the spike time (see Figure 2B), and Bisection, which repeatedly bisects an interval containing the spike. We describe both in Appendix Section D, but primarily use Newton-Raphson due to its fast convergence rate (see Figure 7). A natural question is: what happens if our solvers fail to converge? In our setting, we can guarantee the discovery of spike times due to our ability to bracket the spike time and the unimodality of the voltage function $V(t)$ between input spikes (see Appendix Section D.1), which means that there is only one spike time per interval between input spikes.

**Gradients through the root solver.** A final challenge is computing gradients of the spike time $t^\star$ with respect to the neuron's parameters $\mathbf{p} = (V_0, I_0, V_{\text{th}})$. Rather than differentiating through the iterative solver steps, we use the Implicit Function Theorem (Mostafa, 2018; Chen et al., 2018; Wunderlich & Pehle, 2021) to compute these gradients directly. In particular, we aim to compute the Jacobian $\partial t^\star / \partial \mathbf{p}$. We describe the derivation in Appendix Section D.3.

### 3.3. Translating Spike Trains to Class Logits

Our model's final layer is composed of $N_{\text{cls}}$ weighted leaky integrators (i.e., no spiking non-linearity) (Nowotny et al., 2025), one for each class in the classification task, that accumulate incoming spikes over time, which we directly use as logits in a cross-entropy loss function. We can use

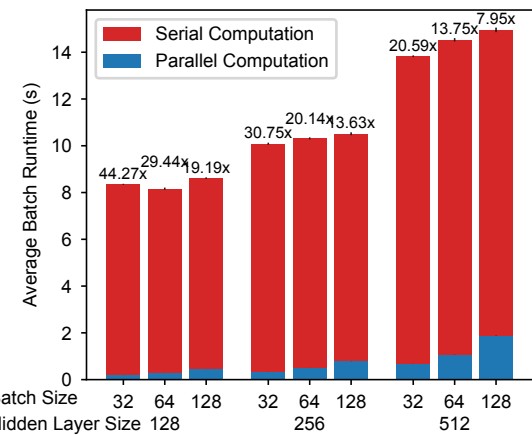

*Figure 8.* Speedup of our parallel SNN implementation over a sequential event-based SNN implementation with exact hard-reset dynamics on the SHD dataset as we vary the hidden dimension and batch size. The model architecture uses 2 feedforward hidden layers and delays. Averages are taken over 100 batches (forward and backward pass) over 3 runs and chunk size is set to 128.

the associative operator *Combine* to implement a Leaky Integrator (LI) neuron, which follows the same dynamics as the LIF neuron, except it does not produce output spikes. Logits are computed via the following integral

$$\int_{t=0}^{t=\tau_{\max}} e^{-t/\tau_{\text{LI}}} V(t) dt. \tag{10}$$

where $\tau_{\max}$ is some prespecified maximum and $\tau_{\text{LI}}$ is the exponential time constant. The intuition behind this variant is that spikes that arrive earlier at the output neurons are weighted more heavily than spikes that arrive later. We provide complete closed-form equations for computing this integral in an *event-based* manner in the Appendix. Furthermore, complete training details, including hyperparameters, spike-count regularizer specification, dataset preprocessing, and other implementation details are provided in the Appendix.

# 4. Experiments

Our experiments probe several research questions: 1) How much speedup can be achieved by processing input spikes in parallel using associative scans while maintaining exact hard-reset dynamics? 2) Can numerical spike time solvers such as Newton-Raphson be used to train SNNs on event-based datasets? 3) How does discretizing spike times into time bins impact classification accuracy on a task requiring sub-millisecond precision?

Our experiments are performed on four event-based datasets, including the Spiking Heidelberg Digits (SHD) dataset (Cramer et al., 2022), the Spiking Speech Commands (SSC) dataset (Cramer et al., 2022), the latency-encoded MNIST

dataset, and the Yin-Yang (YY) dataset (Kriener et al., 2022). Our model architecture is composed of fully connected layers of LIF neurons. The output layer is composed of $N_{\text{cls}}$ leaky integrator neurons, where $N_{\text{cls}}$ is the number of classes in the dataset. Complete training details are provided in the Appendix. We compare our method against a sequential event-based SNN implementation with exact hard-reset dynamics, which consumes input spikes one-by-one and after each spike checks if the neuron will produce an output spike before the next input spike arrives. If so, the neuron produces the output spike and resets its membrane potential to zero before continuing to consume input spikes. We report training speed compared to this serial baseline and explore how the chunk size (i.e., the number of input spikes processed in parallel) impacts training speed. We also compare our method's properties and classification accuracy to those of prior works that use discrete-time approximations, analytical spike time solutions, and surrogate gradients.

## 4.1. Parallel Associative Scan Speedups & Optimal Chunk Size

We measure the training time per batch of our parallel SNN implementation against a sequential/serial event-based SNN implementation with exact hard-reset dynamics. We vary the batch size, number of hidden units, and chunk size (i.e., the number of input spikes processed in parallel) to understand how these parameters impact speedup. Here, we report results on the SHD dataset and report SSC results in Appendix Section F.

In Figure 8, we report that our parallel SNN implementation achieves up to 44x speedup over the sequential baseline. At larger batch sizes and larger hidden dimensions, we still observe large speedups, particularly in absolute terms, where sequential training slows down dramatically. We also find that chunk size plays a significant role in maximizing throughput as seen in Figure 6. We observe that the optimal chunk size is determined by a combination of the number of hidden units and batch size. Larger chunk sizes allow for more parallelism, but also require moving more data—taxing memory bandwidth—and scheduling more work, which can lead to diminishing returns. We find that chunk size 128 works well across a variety of batch sizes and hidden dimensions. Larger chunk sizes may also lead to more wasted computation, since more work is discarded when a neuron spikes early in the chunk. However, we note that this effect is manageable in practice since neurons tend to spike sparsely relative to the number of input spikes they receive. We quantify the percentage of work retained in Appendix Section G.

*Table 1.* Classification accuracy comparison on event-based datasets. Model architectures are defined by number of layers and connectivity pattern (F for feedforward, R for recurrent), number of hidden units (H), and use of delays (D) (e.g., 2F512HD means 2 layers, feedforward connectivity, 512 hidden units, and use of delays). $N$ is the number of spikes processed by a single neuron and $T$ is the number of time steps in the simulation. Our metrics are averaged over 5 random seeds and standard deviation is reported in parentheses.

| Dataset | Model | Exact Gradients | Continuous Spike Times | Parallelized | Compute Depth | Memory | Accuracy |
|---|---|---|---|---|---|---|---|
| MNIST | 1F350H, $\tau_m = 2\tau_s$ (Göltz et al., 2021) | ✓ | ✓ | x | $O(N)$ | $O(N)$ | 97.20 (0.10) |
| | 1F350H (Wunderlich & Pehle, 2021) | ✓ | ✓ | x | $O(N)$ | $O(N)$ | 97.60 (0.10) |
| | 1F350H (Ours) | ✓ | ✓ | ✓ | $O(C \log K)$ | $O(N)$ | 98.04 (0.07) |
| SHD | 2F256HD (Hammouamri et al., 2024) | x | x | x | $O(T)$ | $O(T)$ | 95.07 (0.24) |
| | 2F512HD (Mészáros et al., 2025) | ✓ | x | x | $O(T)$ | $O(N)$ | 93.10 |
| | 2F256HD (Ours) | ✓ | ✓ | ✓ | $O(C \log K)$ | $O(N)$ | 94.77 (0.46) |
| | 2F512HD (Ours) | ✓ | ✓ | ✓ | $O(C \log K)$ | $O(N)$ | 94.96 (0.23) |
| SSC | 2F512HD (Hammouamri et al., 2024) | x | x | x | $O(T)$ | $O(T)$ | 80.69 (0.21) |
| | 2F512HD (Mészáros et al., 2025) | ✓ | x | x | $O(T)$ | $O(N)$ | 76.10 (1.00) |
| | 2F512HD (Ours) | ✓ | ✓ | ✓ | $O(C \log K)$ | $O(N)$ | 77.79 (0.19) |

## 4.2. Classification Accuracy

Table 1 summarizes key attributes of different methods and their classification accuracy on event-based datasets.

**MNIST.** We note the favorable computational scaling properties of our method, which has compute depth $O(C \log K)$ due to our use of parallel associative scans, compared to either $O(N)$ or $O(T)$ for other methods, where $T$ is the number of discrete time steps in the simulation. We also note an improved classification accuracy of 98.04% for the same parameter count compared to prior works that use exact gradients with analytical spike time solutions where the membrane and synaptic time constants are restricted to have the relation $\tau_m = 2\tau_s$ (Göltz et al., 2021).

**SHD and SSC.** We compare against methods that use exact gradients with discrete spike times (Mészáros et al., 2025) and surrogate gradients with a discrete-time grid (Hammouamri et al., 2024) (see Figure 5 for a visualization of how discretization affects the smoothness of the optimization landscape). Compared to Mészáros et al. (2025), we find that our method performs favorably on both the SHD and SSC datasets in terms of classification accuracy, confirming the feasibility of using numerical spike time solvers in SNN training and the potential advantages of continuous-time methods.[2] We also note that our method not only operates over continuous spike times, but also operates over continuous delays, whereas Mészáros et al. (2025) use discrete delays. Notably, our system supports continuous delays *and* multiple spikes per neuron—an important distinction from other recent event-based continuous-time SNNs with delays (Göltz et al., 2025) (see Appendix Figure 11 for observed per-layer spike counts during training).

Our method's accuracy is comparable to that of Hammouamri et al. (2024) on the SHD dataset and lower on the SSC dataset. Hammouamri et al. (2024)'s use of surrogate gradients leads to a loss of machine-precision spike times, $O(T)$ compute/memory (vs. our $O(C \log K)$ compute, $O(N)$ memory), and binned input spike trains. We use full spike trains with no loss of temporal resolution. We note that current standard benchmarks (SHD, SSC) are solvable via rate-coding, which favors the smoothing properties of surrogate gradients (Ma et al., 2025). While benchmarks requiring strict spike-timing codes are emerging, to our knowledge there is not yet a widely adopted community-standard large-scale benchmark whose performance robustly depends on such codes. Our method provides the necessary approach to train on such future datasets efficiently.

## 4.3. Impact of Discretization on Temporal Coding Tasks

To demonstrate the limitations of discrete-time methods on temporal coding tasks, we created a barn owl-inspired inter-aural time difference (ITD) task using the Yin-Yang dataset (Kriener et al., 2022). Each sample consists of 5 input spikes arriving within 2ms, where spike times encode the $(x, y)$ position on the Yin-Yang symbol. The 3-way classification (yin, yang, dot) requires sub-millisecond discrimination—analogous to barn owl sound localization (Carr & Konishi, 1990). We trained our method under different discrete-time constraints to simulate what discrete-time methods would experience. For each time step size $\Delta t$, we quantized all spike times to the nearest multiple of $\Delta t$ during training and testing. Figure 9 shows that accuracy degrades significantly as $\Delta t$ increases, dropping to near-chance levels (33%) at $\Delta t \geq 2$ms. Our method with continuous spike times maintains the highest accuracy. This demonstrates that achieving sub-millisecond precision (e.g., $\Delta t = 0.1$ms) would require discrete-time methods to incur 10x higher memory and computation costs compared to 1ms bins, whereas our

---

[2]Best reported accuracy from Mészáros et al. (2025) on SHD is 93.24 (1.0), though we benchmark comparable architectures.

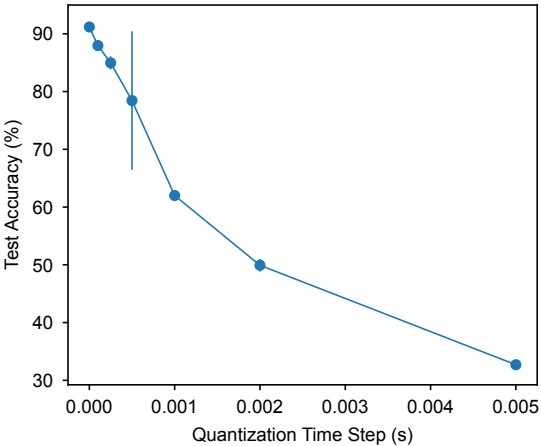

*Figure 9.* Impact of temporal discretization on Yin-Yang classification. We trained our method under different discrete-time constraints by quantizing spike times to multiples of $\Delta t$ during training and testing. Accuracy degrades significantly as $\Delta t$ increases, dropping to near-chance (33%) at $\Delta t \geq 2$ms. Continuous spike times ($\Delta t \to 0$, our standard method) maintain the highest accuracy, demonstrating that discrete-time methods would require $\Delta t \ll 1$ms—incurring substantial computational costs—to preserve temporal information.

continuous-time approach achieves this precision without such penalties.

## 5. Discussion

This work opens up several avenues for future research. One potential direction is to explore the application of our method to more complex SNN architectures, such as convolutional or recurrent networks. Recurrence in particular poses interesting challenges for parallel processing on modern hardware, such as GPUs, because the input queues to each neuron are dynamic. While data structures such as heaps/priority queues are sensible choices to consider, they are very indexing- and memory-access-intensive, which may make them ill-suited for GPU architectures (Engelken, 2023; Landsmeer et al., 2025). Additionally, now that we have demonstrated the effectiveness of using numerical spike time solvers in SNN training, we can explore a wider range of neuron models, which may lead to improved performance on event-based tasks. Finally, evaluating the impact of training with precise spike times on downstream deployment to temporally precise neuromorphic hardware is an exciting direction and is left for future work.

This paper presents a novel combination of methods that unlocks flexible and efficient training of temporally precise spiking neural networks on modern parallel hardware, specifically via parallel associative scans—while maintaining hard-reset dynamics—and differentiable spike time solvers. We demonstrate significant speedups in training

time over sequential event-based SNN approaches. We also demonstrate that our numerical spike time solver is capable of machine-precision spike times, which enables investigations into how the brain might leverage spike timing and ordering for computation, compatibility with high resolution sensors and neuromorphic hardware, and computing on an event-by-event basis, where memory and compute scale with the number of spikes processed rather than the number of discrete time steps simulated. Furthermore, we enable continuous spike times while removing the restriction that the neuron model must have an analytical spike time solution. Our method achieves strong classification accuracy on event-based datasets compared to prior works that use discrete-time approximations, indicating the viability of using numerical spike time solvers in SNN training.

## Software and Data

Our codebase is available at `https://github.com/ToddMorrill/snn-bullet-trains`. The datasets used in this work are publicly available at the following URLs: SHD and SSC datasets `https://tonic.readthedocs.io/en/latest/datasets.html`, and MNIST dataset `https://docs.pytorch.org/vision/main/generated/torchvision.datasets.MNIST.html`.

## Acknowledgements

We thank Richard Zemel and Benjamin Eyre for helpful discussions and feedback on the manuscript. This work was supported by funds provided by the National Science Foundation and by DoD OUSD (R&E) under Cooperative Agreement PHY-2229929 (The NSF AI Institute for Artificial and Natural Intelligence), the US National Institutes of Health Grant S10OD028632-01, and the Schmidt Foundation.

## Impact Statement

This paper presents work whose goal is to advance the field of Machine Learning. There are many potential societal consequences of our work, none of which we feel must be specifically highlighted here.

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

## A. Solving the LIF Neuron ODE System

Here we solve the LIF neuron ordinary differential equation (ODE) system, which provides closed-form expressions for $V(t)$ and $I(t)$, which are useful in later calculations. We are given

$$\tau_m \frac{dV}{dt} = -V(t) + I(t), \quad V(0) = V_0 \tag{11}$$

$$\tau_s \frac{dI}{dt} = -I(t), \quad I(0) = I_0 \tag{12}$$

where $V_0$ and $I_0$ are the initial membrane potential and synaptic input current at time $t = 0$. We can solve equation 12 through a separation of variables. We have

$$-\tau_s \int_{I(t'=0)}^{I(t'=t)} \frac{1}{I(t')} dI = \int_{t'=0}^{t'=t} dt' \tag{13}$$

$$\log(I(t')) \Big|_{t'=0}^{t'=t} = -\frac{t}{\tau_s} \tag{14}$$

$$\log \left( \frac{I(t)}{I_0} \right) = -\frac{t}{\tau_s} \tag{15}$$

$$I(t) = I_0 e^{-t/\tau_s}. \tag{16}$$

Next, we solve for $V(t)$ through the use of an integration factor $e^{t/\tau_m}$. We have

$$\frac{dV}{dt} e^{t/\tau_m} = \frac{1}{\tau_m} I_0 e^{-t/\tau_s} e^{t/\tau_m} - \frac{1}{\tau_m} V(t) e^{t/\tau_m} \qquad \text{(sub. in eq. 16)} \tag{17}$$

$$\frac{d}{dt} \left( V(t) e^{t/\tau_m} \right) = \frac{1}{\tau_m} I_0 e^{-t/\tau_s} e^{t/\tau_m} \tag{18}$$

where $\frac{d}{dt} \left( V(t) e^{t/\tau_m} \right) = \frac{dV}{dt} e^{t/\tau_m} + \frac{V(t) e^{t/\tau_m}}{\tau_m}$ is used in equation 18 and holds by the chain rule.

Let $A = \frac{1}{\tau_s} - \frac{1}{\tau_m}$. Integrating both sides, we have

$$\int_{t'=0}^{t'=t} \frac{d}{dt'} \left( V(t') e^{t'/\tau_m} \right) dt' = \frac{I_0}{\tau_m} \int_{t'=0}^{t'=t} e^{-t'A} dt' \tag{19}$$

$$V(t) e^{t/\tau_m} - V(0) = -\frac{I_0}{\tau_m A} e^{-tA} + \frac{I_0}{\tau_m A} \tag{20}$$

$$V(t) e^{t/\tau_m} - V(0) = I_0 \frac{\tau_s}{\tau_m - \tau_s} [1 - e^{-(t/\tau_s - t/\tau_m)}]. \tag{21}$$

Finally, multiplying by $e^{-t/\tau_m}$, the inverse of the integration factor, we have

$$V(t) = V_0 e^{-t/\tau_m} + I_0 \frac{\tau_s}{\tau_m - \tau_s} [e^{-t/\tau_m} - e^{-t/\tau_s}]. \tag{22}$$

Note that we assume $\tau_m \neq \tau_s$ to avoid division by 0. For the $\tau_m = \tau_s$ case, the system can be solved analytically (see Göltz et al. (2021) for details), though our associative scan method still applies. The complete solution to the system is given as

$$V(t) = V_0 e^{-t/\tau_m} + I_0 \frac{\tau_s}{\tau_m - \tau_s} [e^{-t/\tau_m} - e^{-t/\tau_s}] \tag{23}$$

$$I(t) = I_0 e^{-t/\tau_s}. \tag{24}$$

## B. Associative Scan

Recall that we propose a form for the associative *Combine* operator as

$$Combine((M_2, \mathbf{b}_2), (M_1, \mathbf{b}_1)) = (M_2 M_1, M_2 \mathbf{b}_1 + \mathbf{b}_2). \tag{25}$$

**Lemma B.1.** *Let* $\mathbf{s}_1 = M_1\mathbf{s}_0 + \mathbf{b}_1$ *and* $\mathbf{s}_2 = M_2\mathbf{s}_1 + \mathbf{b}_2$. *Then the operator* Combine *defined as* $Combine((M_2, \mathbf{b}_2), (M_1, \mathbf{b}_1)) = (M_2M_1, M_2\mathbf{b}_1 + \mathbf{b}_2)$ *is associative, i.e.,* $Combine(Combine(a, b), c) = Combine(a, Combine(b, c))$ *for any* $a, b, c$.

*Proof.* We want to show that $Combine(Combine(a, b), c) = Combine(a, Combine(b, c))$. Expanding the left-hand-side, we have

$$Combine((M_aM_b, M_a\mathbf{b}_b + \mathbf{b}_a), (M_c, \mathbf{b}_c)) = (M_aM_bM_c, M_aM_b\mathbf{b}_c + M_a\mathbf{b}_b + \mathbf{b}_a) \tag{26}$$

and expanding the right-hand-side, we have

$$Combine((M_a, \mathbf{b}_a), (M_bM_c, M_b\mathbf{b}_c + \mathbf{b}_b)) = (M_aM_bM_c, M_aM_b\mathbf{b}_c + M_a\mathbf{b}_b + \mathbf{b}_a), \tag{27}$$

which proves their equivalence and allows us to conclude that *Combine* is an associative operator. □

We note that our use of parallel associative scans is facilitated by the ability to express the inter-spike dynamics as affine maps. This holds for any neuron model with linear dynamics. Specifically, any model where the state evolution between spikes follows

$$\frac{d\mathbf{s}}{dt} = A\mathbf{s}(t) + \mathbf{c}(t) \tag{28}$$

where $A$ is a constant matrix and $\mathbf{c}(t)$ is a state-independent vector, admits a closed-form transition operator that is independent of the initial state $\mathbf{s}(0)$. This allows the temporal evolution to be parallelized via associativity.

## C. Event-Based Leaky Integrator

We now derive event-based implementations of the unweighted and weighted leaky integrator (LI) neuron models (Nowotny et al., 2025), which is used in the final layer of our SNN to translate output spike trains into class logits. Recall that a LI neuron computes the integral

$$\int_{t=0}^{t=\tau_{\max}} V(t)dt \tag{29}$$

where $t = 0$ is the start of the integration period and $\tau_{\max}$ is some prespecified maximum (e.g., 2 seconds). We break the integral into intervals between consecutive spikes and make use of the *Combine* operation to obtain the voltage value at the left and right endpoints of the integrals

$$\sum_{i=0}^{N-1} \int_{t_i}^{t_{i+1}} V(t)dt \tag{30}$$

where $i$ indexes the input spikes. Expanding the integral between spike times $t_i$ and $t_{i+1}$, we have

$$\int_{t_i}^{t_{i+1}} V(t)dt = \int_{t_i}^{t_{i+1}} V_{t_i}e^{-(t-t_i)/\tau_m} + I_{t_i}\frac{\tau_s}{\tau_m - \tau_s}[e^{-(t-t_i)/\tau_m} - e^{-(t-t_i)/\tau_s}]dt \tag{31}$$

$$= V_{t_i}\int_{t_i}^{t_{i+1}} e^{-(t-t_i)/\tau_m}dt \tag{32}$$

$$+ I_{t_i}\frac{\tau_s}{\tau_m - \tau_s}\left[\int_{t_i}^{t_{i+1}} e^{-(t-t_i)/\tau_m}dt - \int_{t_i}^{t_{i+1}} e^{-(t-t_i)/\tau_s}dt\right] \tag{33}$$

$$= V_{t_i}\tau_m(1 - e^{-\Delta t/\tau_m}) + I_{t_i}\frac{\tau_s}{\tau_m - \tau_s}[\tau_m(1 - e^{-\Delta t/\tau_m}) - \tau_s(1 - e^{-\Delta t/\tau_s})], \tag{34}$$

where we've denoted the time between spikes as $\Delta t = t_{i+1} - t_i$. A successful variant of the LI neuron—and the one we use in this work—is the exponentially weighted LI neuron (Nowotny et al., 2025), which is given as

$$\int_{t=0}^{t=\tau_{\max}} e^{-t/\tau_{LI}}V(t)dt, \tag{35}$$

where the analogous closed-form spike-to-spike voltage integral (cf. equation 34) is

$$\int_{t_i}^{t_{i+1}} e^{-t/\tau_{\text{LI}}} V(t) dt = (V_{t_i} + \gamma) e^{t_i/\tau_m} T_m (e^{-t_i/T_m} - e^{-t_{i+1}/T_m}) \tag{36}$$

$$- \gamma e^{t_i/\tau_s} T_s (e^{-t_i/T_s} - e^{-t_{i+1}/T_s}) \tag{37}$$

where $T_m = \frac{\tau_{\text{LI}} \tau_m}{\tau_{\text{LI}} + \tau_m}$ and $T_s = \frac{\tau_{\text{LI}} \tau_s}{\tau_{\text{LI}} + \tau_s}$ and $\gamma = I_{t_i} \frac{\tau_s}{\tau_m - \tau_s}$.

We found that using the weighted leaky integrator directly resulted in outputs with large absolute values, which caused saturation in the softmax operation used in the cross-entropy loss. We explored four solutions to this problem: 1) applying a fixed logit temperature multiplier before the softmax, 2) normalizing the output logits to have zero mean and unit variance (Layer Normalization), 3) normalizing the output logits by the unweighted leaky integrator output (i.e., dividing by Equation 34), and 4) using AdamW (Loshchilov & Hutter, 2018) optimizer with weight decay on the output layer weights. We found that applying a fixed logit temperature multiplier worked best in practice and used this method in all experiments.

## D. Numerical Solvers

We now provide additional details on our numerical spike time solvers.

### D.1. Newton-Raphson Spike Time Solver

The Newton-Raphson update rule for the predicted spike time is given by

$$t_{m+1} = t_m - \frac{R(\mathbf{p}, t_m)}{\frac{\partial R}{\partial t}(\mathbf{p}, t_m)} \tag{38}$$

where $m$ indexes the procedure's iteration. Given a sufficiently close initial guess, Newton-Raphson solvers converge quadratically (i.e., the distance between the current guess and the true spike time, denoted $\epsilon$, has the relation $\epsilon_{m+1} = O(\epsilon_m^2)$) (Press et al., 2007). We initialize this iterative procedure by setting $t_0 = \frac{t_{V_{\max}} - t_{\text{now}}}{2}$, or in other words the midpoint between the current time ($t_{\text{now}} = 0$) and the time that the neuron reaches maximum voltage—note these are relative times, not absolute times. $t_{V_{\max}}$ can be computed analytically, which we describe below.

We now show how $t_{V_{\max}}$ can be computed. A maximum occurs when the condition $\frac{dV}{dt} = 0$ is satisfied, which implies $V(t_{V_{\max}}) = I(t_{V_{\max}})$ by equation 11, which we solve for $t_{V_{\max}}$ to get

$$t_{V_{\max}} = \frac{\tau_m \tau_s}{\tau_m - \tau_s} \log \left( \frac{I_0 \tau_m}{V_0(\tau_m - \tau_s) + I_0 \tau_s} \right). \tag{39}$$

We work with neuron models where $\tau_m > \tau_s$, so the maximum occurs at a positive time when the argument to the $\log$ function is greater than 1, or more specifically, when

$$\frac{I_0 \tau_m}{V_0(\tau_m - \tau_s) + I_0 \tau_s} > 1 \implies I_0 > V_0. \tag{40}$$

We defined unit tests to verify the correctness and numerical precision of our Newton-Raphson spike time solver against analytically computed spike times using the neuron model specified by Göltz et al. (2021) with $\tau_m = 2\tau_s$ ($\tau_m = 20$ms and $\tau_s = 10$ms). We vary the synaptic weight from very small values that cause late spikes to very large values that cause early spikes and find that our implementation recovers ground truth spikes to within $10^{-7}$ seconds of the analytical solution with 14 iterations, even in the worst-case "just grazing" scenarios where the neuron barely reaches threshold.

A natural question is: what happens if Newton-Raphson fails to converge? In our setting, it does not. We 1) only invoke the solver on intervals that our analytic checks certify contain a spike, and 2) initialize in a conservative region where the voltage evolution is well-behaved, and 3) clamp the output time to the search interval on each iteration. Concretely, for each candidate interval $[t_{\text{now}}, t_{\text{next}}]$ we use the analytic condition $V(t_{V_{\max}}) \geq V_{\text{th}}$ (with $t_{V_{\max}} \in [t_{\text{now}}, t_{\text{next}}]$) or $V(t_{\text{next}}) \geq V_{\text{th}}$ to guarantee existence of a threshold crossing. We then initialize the Newton iteration at the midpoint between the left endpoint ($t_{\text{now}} = 0$) and the maximum-voltage time,

$$t_0 = \frac{t_{V_{\max}} - t_{\text{now}}}{2}, \tag{41}$$

which lies inside an interval where the membrane potential is monotone increasing. For our current-based LIF model (Equation 22) the membrane potential $V(t)$ is a linear combination of two distinct exponentials, $e^{-t/\tau_m}$ and $e^{-t/\tau_s}$. Consequently, its time derivative $\frac{dV}{dt}$ also takes the form of a sum of two exponentials:

$$\frac{dV}{dt} = c_1 e^{-t/\tau_m} + c_2 e^{-t/\tau_s} \tag{42}$$

Setting $\frac{dV}{dt} = 0$ to find critical points yields the equation:

$$e^{t(\frac{1}{\tau_s} - \frac{1}{\tau_m})} = -\frac{c_2}{c_1} \tag{43}$$

Since the exponential function on the left-hand side is strictly monotonic (given $\tau_m \neq \tau_s$), this equation has at most one solution for $t$. Therefore, $\frac{dV}{dt}$ changes sign at most once. Given that $V(t)$ decays to 0 as $t \rightarrow \infty$, if a peak exists at $t_{V_{\max}} > t_{\text{now}}$, the function must be strictly increasing on $[t_{\text{now}}, t_{V_{\max}}]$ and strictly decreasing thereafter. This unimodality guarantees that the residual $R(t) = V(t) - V_{\text{th}}$ has a unique root within the search interval, ensuring non-oscillatory convergence of the Newton solver.

Operationally, we run a fixed 14 Newton iterations, which our tests show suffices to reach float32 machine-precision over the entire weight range (including the grazing case). Our method operates at machine-precision for single-precision floating point numbers (i.e., float32), which follows from the fact that our implementation must compute $V(t) - V_{\text{th}}$. In our work $V_{\text{th}} = 1.0$, and at this scale, the next representable float32 number is 1.0000001192092896. So, for example, once the numerical root solvers find $t$ such that $V(t)$ is equal to this number, further iterations will return the same spike time. Therefore, our method is able to recover spike times to machine-precision permitted by the root finding procedure in float32. We also note that the term $-\left(\frac{\partial R}{\partial t}\right)^{-1}$ in Equation 45 can result in spike time divergence if the derivative is very close to zero. To prevent this issue, we employ a Solver Regularization term—consistent with other exact gradient methods that address diverging spike times (Mostafa, 2018; Göltz et al., 2021)—specified in Table 2 that limits the minimum absolute value of the derivative.

### D.2. Bisection Solver

Here, we define the Bisection solver's algorithm. The Bisection method requires 20 iterations to achieve similar numerical precision as the Newton-Raphson method in our tests.

---

**Algorithm 1** Bisection Method for Spike Time

---

**Input:** LIF parameters $\theta$, initial state $\mathbf{s}_0 = (V_0, I_0)$, time of maximum voltage $t_{V_{\max}}$, maximum iterations $max\_iter$
Define function $R(t) = V(t) - V_{\text{th}}$ where $V(t)$ is given by equation 22
Initialize $t_{low} = 0.0$, $t_{high} = t_{V_{\max}}$
**for** $i = 1$ **to** $max\_iter$ **do**
 $t_{mid} = \frac{t_{low} + t_{high}}{2.0}$
 **if** $R(t_{mid}) > 0$ **then**
  $t_{high} = t_{mid}$
 **else**
  $t_{low} = t_{mid}$
 **end if**
**end for**
**Output:** Estimated spike time $t_{solution} = \frac{t_{low} + t_{high}}{2.0}$

---

Note that the Bisection method requires that you can bracket the root, i.e., find an interval $[t_{low}, t_{high}]$ such that $R(t_{low})$ and $R(t_{high})$ have opposite signs. We are able to do this by setting $t_{low} = 0$ and $t_{high} = t_{V_{\max}}$. Again, recall that we can evaluate $V(t_{V_{\max}})$ to determine if it is greater than $V_{\text{th}}$, in which case we satisfy the requirement of a valid bracketing that $R(t_{low})$ and $R(t_{high})$ have opposite signs. The Bisection solver has a slower convergence rate compared to the Newton-Raphson method with $\epsilon_{m+1} = \frac{1}{2}\epsilon_m$. In contrast to the Newton-Raphson method, the Bisection method does not require knowledge of the derivative of $R(t)$. Both methods still require a procedure for evaluating $V(t)$ given the initial state $\mathbf{s}_0$ and time $t$.

### D.3. Implicit Function Theorem Gradients

Recall that our aim is to compute the gradient of the output spike time $t^\star$ with respect to the parameters $\mathbf{p}$ that govern the neuron's dynamics. The spike time is implicitly defined as the root of the function

$$R(\mathbf{p}, t(\mathbf{p})) = V(\mathbf{p}, t(\mathbf{p})) - V_{\text{th}}. \tag{44}$$

In order for $t(\mathbf{p})$ to be well-defined as a function locally in a neighborhood around the point $(\mathbf{p}, t^\star)$, it must be the case that $\partial R/\partial t|_{(\mathbf{p}, t^\star)} \neq 0$. If this condition is satisfied, then we can solve for the gradient of the output spike time with respect to the parameters as follows. Since $R(\mathbf{p}, t(\mathbf{p})) = 0$ holds for all $\mathbf{p}$ in a valid neighborhood, its gradient with respect to $\mathbf{p}$ must also be zero. Applying the multivariate chain rule gives

$$\nabla_{\mathbf{p}} R(\mathbf{p}, t(\mathbf{p})) = \frac{\partial R}{\partial \mathbf{p}} + \frac{\partial R}{\partial t} \frac{\partial t}{\partial \mathbf{p}} = 0$$

$$\frac{\partial t}{\partial \mathbf{p}} = -\left(\frac{\partial R}{\partial t}\right)^{-1} \cdot \frac{\partial R}{\partial \mathbf{p}}. \tag{45}$$

Recall that $\frac{\partial R}{\partial t} = \frac{\partial V}{\partial t}$. This highlights a boundary case when $\frac{\partial V}{\partial t} = 0$, which is when the spike time diverges to infinity. Also note that equation 45 defines a Vector-Jacobian Product (VJP) rule in `JAX` for any root solver.

## E. Implementation Details

We now describe key implementation details of our parallel SNN implementation. Our models are feedforward fully connected SNNs composed of LIF neurons with hard-reset dynamics governed by the ODE system defined in equations 11 and 12. Our loss function has two components: 1) a classification loss, and 2) a spike-count regularizer. The classification loss is a cross-entropy loss on the output logits produced by the weighted leaky integrator output as defined in equation 35 of Appendix Section C. The spike-count regularizer pulls spiking activity towards a target spike count. In particular, our complete loss function is given as

$$\mathcal{L} = \mathcal{L}_{CE}(\hat{y}, y) + \lambda_{\text{reg}} \mathcal{L}_{\text{reg}} + \lambda_{\text{over-reg}} \mathcal{L}_{\text{over-reg}} \tag{46}$$

where $\mathcal{L}_{CE}$ is the cross-entropy loss between the predicted class logits $\hat{y}$ and the ground truth class $y$, $\mathcal{L}_{\text{reg}}$ is the spike-count regularization loss (specified below), $\lambda_{\text{reg}}$ is a hyperparameter that controls the strength of the regularization, and $\lambda_{\text{over-reg}}$ and $\mathcal{L}_{\text{over-reg}}$ control overactive neurons (described below). We now describe the spike-count regularization terms in detail.

### E.1. Spike Count Regularization

To encourage a biologically plausible firing activity, we introduce a regularization term to the loss function. As the spike count itself is non-differentiable, we optimize a differentiable proxy based on the membrane potential. The core idea is that if a neuron is over-active, we penalize the maximum voltage ($V_{\text{max}}$) in the intervals where it spiked, pushing the voltage trajectory down. Conversely, if a neuron is under-active, we penalize the $V_{\text{max}}$ in the intervals where it remained silent, pushing the trajectory up towards the firing threshold.

Let $c_{\text{target}}$ be the target spike count per trial. For a single neuron $i$ and a single data instance $k$ from a batch of size $N_{\text{batch}}$, let $c_i^{(k)}$ be its measured spike count. We consider the $M_i^{(k)}$ intervals defined by the input spike train for this instance. For each interval $j \in \{1, \ldots, M_i^{(k)}\}$, we can analytically compute the maximum voltage potential, denoted $V_{\text{max}}^{(i,j,k)}$.

We partition the intervals into two sets for each neuron and each instance:

- $\mathcal{S}_i^{(k)} = \{j \mid \text{neuron } i \text{ spiked in interval } j \text{ on instance } k\}$

- $\bar{\mathcal{S}}_i^{(k)} = \{j \mid \text{neuron } i \text{ remained silent in interval } j \text{ on instance } k\}$

Note that the measured spike count for the instance is $c_i^{(k)} = |\mathcal{S}_i^{(k)}|$.

To regulate the firing, we first define two per-instance auxiliary loss terms. For the over-firing case, the loss penalizes any voltage maximum that exceeds the threshold $V_{\text{th}}$. For the under-firing case, the loss penalizes any voltage maximum that

falls below the threshold.

$$\mathcal{L}_{\text{under}}^{(i,k)} = \frac{1}{|\bar{\mathcal{S}}_i^{(k)}|} \sum_{j \in \bar{\mathcal{S}}_i^{(k)}} \text{ReLU}\left(V_{\text{th}} - V_{\text{max}}^{(i,j,k)}\right) \tag{47}$$

$$\mathcal{L}_{\text{over}}^{(i,k)} = \frac{1}{|\mathcal{S}_i^{(k)}|} \sum_{j \in \mathcal{S}_i^{(k)}} \text{ReLU}\left(V_{\text{max}}^{(i,j,k)} - V_{\text{th}}\right) \tag{48}$$

where $\text{ReLU}(x) = \max(0, x)$ and is short for Rectified Linear Unit (ReLU). We then average these per neuron loss terms over the batch, alongside the spike count:

$$\bar{c}_i = \frac{1}{N_{\text{batch}}} \sum_{k=1}^{N_{\text{batch}}} c_i^{(k)} \tag{49}$$

$$\bar{\mathcal{L}}_{\text{under}}^{(i)} = \frac{1}{N_{\text{batch}}} \sum_{k=1}^{N_{\text{batch}}} \mathcal{L}_{\text{under}}^{(i,k)} \tag{50}$$

$$\bar{\mathcal{L}}_{\text{over}}^{(i)} = \frac{1}{N_{\text{batch}}} \sum_{k=1}^{N_{\text{batch}}} \mathcal{L}_{\text{over}}^{(i,k)} \tag{51}$$

The total regularization loss for neuron $i$ is a weighted sum of these batch-averaged terms, activated based on whether the neuron's average spike count is above or below the target.

$$\mathcal{L}_{\text{reg}}^{(i)} = \text{ReLU}\left(1 - \frac{\bar{c}_i}{c_{\text{target}}}\right) \bar{\mathcal{L}}_{\text{under}}^{(i)} + \text{ReLU}\left(\frac{\bar{c}_i}{c_{\text{target}}} - 1\right) \bar{\mathcal{L}}_{\text{over}}^{(i)} \tag{52}$$

The final regularization loss is the average over all $n$ neurons in the entire network: $\mathcal{L}_{\text{reg}} = \frac{1}{n} \sum_{i=1}^{n} \mathcal{L}_{\text{reg}}^{(i)}$.

**Regularizing overactive neurons.** Even with the above regularizers, some neurons can become overactive outliers, which results in neurons that don't consume their full input queue and trigger the output spike cap described in Section E.2. To mitigate this issue, we apply an additional regularizer that penalizes *individual* neurons that exceed the target spike count, rather than just penalizing the average spike count across the batch. Concretely, we add a term to the loss that penalizes any neuron $i$ in instance $k$ whose spike count exceeds the target:

$$\mathcal{L}_{\text{over-reg-individual}}^{(i,k)} = \text{ReLU}\left(\frac{c_i^{(k)}}{c_{\text{target}}} - 1\right) \mathcal{L}_{\text{over}}^{(i,k)}. \tag{53}$$

We then average across neurons to get the final individual over-firing regularization term:

$$\mathcal{L}_{\text{over-reg}} = \frac{1}{n} \sum_{i=1}^{n} \sum_{k=1}^{N_{\text{batch}}} \mathcal{L}_{\text{over-reg-individual}}^{(i,k)}. \tag{54}$$

This term is added to the overall loss with its own hyperparameter weight $\lambda_{\text{over-reg}}$ that controls the strength of this regularization. We find that this additional term is effective at preventing outlier neurons that would trigger the output spike cap, which in turn allows us to consume the full input queue without needing to set a high cap that would increase runtimes.

### E.2. Event Queues

Working with precise spike times deviates from the common practice of discretizing time into fixed steps and using a ring buffer to manage synaptic delays (Landsmeer et al., 2025). Instead, we maintain sorted event queues for each neuron that store incoming spikes along with their arrival times. When a neuron processes its queue, it retrieves spikes in chronological order, ensuring accurate temporal dynamics. We process events layerwise, meaning all neurons in a layer process their queues, generate spikes, and then the resulting spikes are sorted into the queues of the next layer. This approach preserves the temporal order of spikes across layers and allows for our chunked parallel processing method to be applied effectively.

It is also important to elucidate how we handle spiking within chunks. Within a given chunk, we only ever "commit" a single output spike. After the parallel scan computes the per-interval analytic spike indicators (Section 3.2.2), we select the first interval whose indicator is `True` (implemented as an `argmax` over a boolean array), solve for the corresponding spike time, hard-reset the neuron, and discard any within-chunk work after that time. This is not an approximation—it matches the semantics of a serial event simulator, which also stops processing input events once a neuron fires and then resumes from the post-reset state. If a neuron spikes again immediately after reset, this is handled in the next iteration in exactly the same way—the new "current time" becomes the just-emitted spike time, and we again test for a spike between that time and the next pending input spike time. Thus, multiple spikes per neuron are represented as multiple iterations over chunks/intervals. Each chunk iteration advances the state to the first output spike (or completely consumes the chunk if no spike occurs), preserving perfect temporal fidelity.

Note that in order to compile the `jax.jit` model function, we must prespecify the number of chunks that each layer will process. We use the following heuristic for determining the number of chunks per layer. First, we observe the maximum number of input spikes the input layer will receive, which we denote as $N_{\text{input}}$. Next, we specify a desired chunk size $K$ and maximum number of output spikes permitted per neuron in each layer $S_{\text{max}}^{(i)}$ for layer $i$. Then for the input layer, we set the number of chunks to $\lceil N_{\text{input}}/K \rceil + S_{\text{max}}^{(0)}$ so that each chunk processes at most $K$ input spikes plus an allowance for output spikes that may be generated within the chunk. For subsequent layers $i > 0$, we set the number of chunks to be $\lceil (S_{\text{max}}^{(i-1)} \cdot n^{(i-1)}/K) + S_{\text{max}}^{(i)} \rceil$ where $n^{(i-1)}$ is the number of neurons in the previous layer.

To bound memory and runtimes, we cap the number of retained output spikes per neuron (e.g., 42 for SHD/SSC). When spiking activity is unusually high, this cap can become active, which may prevent complete consumption of the input queue within the compiled compute budget. In practice, with $S_{\text{max}} = 42$ we consume approximately 100% of input spikes on average in SSC layer 1 and layer 2 (see Appendix Figure 11), and we regularize spike activity to keep this fraction high while preserving accuracy. Increasing the cap improves consumption at the cost of runtime: for example, raising from 28 to 35 in layer 1 increased epoch time by ~18% due to additional scan steps and larger downstream queues, and empirically increased the consumed-spike fraction. When the cap does not activate (or as $S_{\text{max}}$ and the compute budget are increased), our implementation recovers the exact event-based hard-reset forward dynamics and the corresponding Implicit Function Theorem/EventProp (Wunderlich & Pehle, 2021) gradients. The cap is therefore a practical knob for trading memory/throughput against worst-case spiking activity, orthogonal to our core contributions (parallel scans and continuous-time spike solvers). Future work can explore adaptive strategies that tune this cap online based on observed activity. Notably, we designed a regularizer (see Section E.1) that penalizes individual outlier neurons that trigger the cap, which is a more targeted way to mitigate this issue.

### E.3. Sequential Baseline

While the main text describes our parallel SNN implementation, we also developed a sequential event-based SNN implementation with exact hard-reset dynamics to serve as a baseline for speedup comparisons. In this sequential implementation, each neuron processes its input spikes one at a time in chronological order, updating its state and generating output spikes as needed. In order to make this baseline even more competitive, we use the `jax.lax.scan` primitive to efficiently loop over input spikes, and check for output spikes (note all `jax` methods we use are JIT-compiled). We also only call the solver when a spike is detected between input spikes using our lightweight analytical checks, which is an optimization over a naive implementation that calls the solver at every input spike. This sequential implementation serves as a benchmark to highlight the efficiency gains achieved by our parallel SNN approach.

### E.4. Dataset Preprocessing

**Yin-Yang.**   The Yin-Yang dataset (Kriener et al., 2022) is a 3-way classification dataset with 5 input spikes. The first two spike times are defined by the point $(x_1, x_2)$ in the 2d plane on the Yin-Yang symbol. $t_{\text{late}}$ is a user-defined time parameter that governs the latest time at which a spike can occur. To symmetrize the input, spikes $(t_{\text{late}} - x_1, t_{\text{late}} - x_2)$ are added. Finally, an optional fifth spike, a bias spike, can be added to serve as both a reference time point and to boost neuronal activity levels in downstream neurons.

For the discretization experiment in Section 4.3, we trained separate models (5 random seeds) at different time step sizes $\Delta t \in \{0, 0.1, 0.25, 0.5, 1.0, 2.0, 5.0\}$ms, where $\Delta t = 0$ represents continuous spike times (our standard method) and $\Delta t > 0$ represents discrete-time constraints that round all spike times to the nearest multiple of $\Delta t$.

**MNIST.**   We work with a latency-encoded representation of MNIST, which has 784 input channels, each with up to 1 input spike, corresponding to the 784 pixels from MNIST digit images. We linearly map pixels in the range $[1, 255]$ to $[t_{\text{late}}, 0]$ so that brighter pixels arrive earlier. $t_{\text{late}}$ is set to $0.030$. Pixels with a value of 0 are ignored. As a means of data augmentation during training, we apply spike dropout at a rate randomly sampled from $(0, 0.2)$ and time jittering to the input spikes by sampling from a zero-mean Gaussian with standard deviation set to $0.003$. We set the maximum number of input spikes to $N_{\text{input}} = 784$, from which we derive the number of chunks for the input layer as described in Section E.2.

**SHD.**   Cramer et al. (2022) introduce the SHD dataset, which is a 20-way classification task, where the 20 classes are spike representations of the spoken words 0 through 9 in English and German. There are 700 input channels each with multiple spikes. We truncate the input spike trains to 1 second and apply the data augmentations described in Nowotny et al. (2025), namely randomly shifting the input channels in the range $[-40, +40]$ and blending pairs of input spike trains from the same class. We set the maximum number of input spikes to $N_{\text{input}} = 14,000$, from which we derive the number of chunks for the input layer as described in Section E.2.

**SSC.**   The SSC dataset (Cramer et al., 2022) is a 35-way classification task, similar to SHD, where the 35 classes are spike representations of spoken words such as "yes", "no", "up", "down", etc. There are 700 input channels each with multiple spikes. We apply no data augmentation to this dataset. We set the maximum number of input spikes to $N_{\text{input}} = 14,000$, from which we derive the number of chunks for the input layer as described in Section E.2.

### E.4.1. WEIGHT INITIALIZATION

In this work, the initial network weights are drawn from a normal distribution layerwise, denoted $\mathcal{N}(\mu^{(i)}, \sigma_i^2)$ where $i$ denotes the layer index and $\mu^{(i)}$ and $\sigma_i$ are defined as

$$\mu^{(i)} = \frac{\alpha_\mu^{(i)} V_{\text{th}}}{n^{(i-1)} \nu^{(0)} \tau_m} \tag{55}$$

$$\sigma_i = \frac{\alpha_\sigma^{(i)} V_{\text{th}}}{\sqrt{n^{(i-1)} \nu^{(0)} \tau_m}} \tag{56}$$

where $\alpha_\mu^{(i)}$ and $\alpha_\sigma^{(i)}$ are layerwise gain parameters, $V_{\text{th}}$ is the membrane threshold, $n^{(i-1)}$ is the number of neurons in the previous layer, $\nu^{(0)}$ is the average input spike rate (spikes per second) per neuron—note the assumption of a constant spike rate across layers—and $\tau_m$ is the membrane time constant. The intuition for these equations is to think of the voltage as a random variable and set the mean and variance of the weights such that the voltage mean is close to threshold and the voltage variance is controlled, specifically for a neuron in layer $i$ we want

$$\mathbb{E}[V^{(i)}] \approx \alpha_\mu^{(i)} V_{\text{th}} = \mu^{(i)} n^{(i-1)} \nu^{(0)} \tau_m \tag{57}$$

$$\text{Var}(V^{(i)}) \approx (\alpha_\sigma^{(i)} V_{\text{th}})^2 = (\sigma_i)^2 n^{(i-1)} \nu^{(0)} \tau_m \tag{58}$$

where we think of $\mu^{(i)} n^{(i-1)} \nu^{(0)} \tau_m$ as the approximate expected voltage contribution from all incoming spikes over the timescale of the membrane time constant, and similarly for the variance. We set $\alpha_\mu^{(i)}$ and $\alpha_\sigma^{(i)}$ to 0.8 for all layers though more work is needed to determine optimal values for these parameters. At the start of training, we estimate $\nu^{(0)}$ by averaging the number of input spikes per neuron over the entire training dataset and dividing by the total input time duration in seconds over 10 batches of training data.

## E.5. Hyperparameters

*Table 2.* System hyperparameters across the 3 datasets used.

| Parameter | Yin-Yang Value | MNIST Value | SHD Value | SSC Value |
|---|---|---|---|---|
| Optimizer | Adam | Adam | Adam | Adam |
| Learning Rate Schedule | Cosine Decay | Exponential | Cosine Decay | Cosine Decay |
| Cosine Decay Steps | 6000 | — | 10000 | 88500 |
| Learning Rate | 0.02 | 0.02 | 0.001 | 0.001 |
| Learning Rate End Value | 0.0001 | 0.001 | 0.0001 | 0.0001 |
| Learning Rate Decay Factor | — | 0.95 | — | — |
| Learning Rate Warmup Steps | 2000 | 500 | 500 | 2000 |
| Delays Learning Rate | 0.02 | 0.02 | 0.001 | 0.001 |
| Delays Learning Rate Warmup Steps | 2000 | 500 | 500 | 2000 |
| Delays Learning Rate Decay Factor | — | 0.95 | — | — |
| Delays Learning Rate End Value | 0.0001 | 0.001 | 0.0001 | 0.0001 |
| Training Epochs | 300 | 300 | 500 | 500 |
| Early Stopping Patience | — | 30 | 100 | 100 |
| Hidden Layer Sizes | [50] | [350] | varies | [512, 512] |
| $\alpha_\mu$ | [0.8] | [0.8] | [0.8, 0.8] | [0.8, 0.8] |
| $\alpha_\sigma$ | [0.8] | [0.8] | [0.8, 0.8] | [0.8, 0.8] |
| Use Delays | [False] | [False] | [True, True, False] | [True, True, False] |
| Delay Activation Function | — | — | Softplus | Softplus |
| $\beta_{SP}$ | — | — | 25.0 | 5.0 |
| Delay Initialization Multiplier | — | — | 0.1 | 0.1 |
| Maximum Percent Missing Spikes | [0.0] | [0.0] | [0.0, 0.0] | [0.0, 0.0] |
| Weight Bump Steps | 2 | 2 | 2 | 2 |
| Weight Bumping Value | 0.02 | 0.02 | 0.02 | 0.02 |
| Output Spikes per Layer | [6] | [6] | [42, 28] | [42, 28] |
| Regularized Spike Count | [3.0] | [3.0] | [14.0, 14.0] | [14.0, 14.0] |
| $\lambda_{reg}$ | 1e-05 | 1e-05 | 0.001 | 0.001 |
| $\lambda_{over\text{-}reg}$ | 0.0 | 0.0 | 5e-05 | 5e-05 |
| $\tau_{LI}$ | 0.01 | 0.5 | 2.0 | 2.0 |
| $\tau_{max}$ | 0.02 | 0.5 | 4.0 | 4.0 |
| Logit Temperature Multiplier | 20.0 | 20.0 | 60.0 | 30.0 |
| Dataset Dropout Probability | 0.0 | 0.2 | 0.0 | 0.0 |
| Time Jitter Standard Deviation | — | 0.003 | 0.0 | 0.0 |
| Number of Blended Samples | — | — | 7644 | 0 |
| Dataset Maximum Time | — | — | — | — |
| Channel Shift Maximum | — | — | 40 | — |
| Batch Size | 128 | 128 | 256 | 256 |
| Solver Method | Newton-Raphson | Newton-Raphson | Newton-Raphson | Newton-Raphson |
| Solver Maximum Iterations | 14 | 14 | 14 | 14 |
| Solver Regularization | 0.01 | 0.01 | 0.01 | 0.01 |
| $\tau_s$ | 0.0005 | 0.005 | 0.005 | 0.005 |
| $\tau_m$ | 0.002 | 0.02 | 0.02 | 0.02 |
| $V_{th}$ | 1.0 | 1.0 | 1.0 | 1.0 |
| $V_{reset}$ | 0.0 | 0.0 | 0.0 | 0.0 |
| $t_{late}$ | 0.002 | 0.03 | — | — |

## E.6. Training Details

All hyperparameters used in our experiments are summarized in Table 2, which we describe in more detail below. We train all models using the Adam optimizer (Kingma & Ba, 2014) and the learning rate schedules defined in Table 2. We employ hyperparameter tuning and early stopping based on validation accuracy, with the exception of the SHD dataset where we use the test accuracy as per prior work (Hammouamri et al., 2024; Schöne et al., 2025). Early stopping is used with a patience specified in Table 2. The number of fully connected hidden layers and their sizes are specified in Table 2, where the number of list elements corresponds to the number of hidden layers and the integer values correspond to the number of neurons in each hidden layer. The use of layerwise delays is similarly specified in Table 2, where a value of True indicates that particular layer uses trainable synaptic delays and False indicates no delays are used. We experimented with several delay activation functions including ReLU, tanh + 1, exponential, and softplus, where the softplus function is defined as $\text{softplus}(x) = (1/\beta_{\text{SP}}) \log(\exp(\beta_{\text{SP}} x) + 1)$, where $\beta_{\text{SP}}$ is a hyperparameter that controls the curvature of the softplus function. We found softplus to perform best in our experiments, where one possible explanation is that it provides a good balance between allowing for small delays with non-zero gradients (unlike ReLU). When delays are used, they are initialized from a uniform distribution in the range $[0, 0.1]$ seconds, where the right endpoint is specified in Table 2 as the Delay Initialization Multiplier. We employ a weight bumping scheme[3] that adds a small constant value (Weight Bumping Value) to all incoming weights of a neuron that has been silent for some number of batches (Weight Bump Steps). We specify the maximum number of output spikes per layer (Output Spikes per Layer)—see their usage in Appendix Section E.2—and a target regularized spike count (Regularized Spike Count). The weighted leaky integrator readout (Equation 35) has two parameters corresponding to the exponential time constant ($\tau_{\text{LI}}$) and the duration of the integration window ($\tau_{\text{max}}$). Our system's time constants are set according to Table 2, where $\tau_m$ is the membrane time constant, $\tau_s$ is the synaptic time constant, $V_{\text{th}}$ is the membrane threshold, $V_{\text{reset}}$ is the reset potential after a spike, and $t_{\text{late}}$ controls the latency encoding of the MNIST dataset. We train on NVIDIA H100 GPUs with 80GB of memory.

---

[3]Similar to the procedure described in (Göltz et al., 2021).

# F. Speedups from Parallel Spike Processing

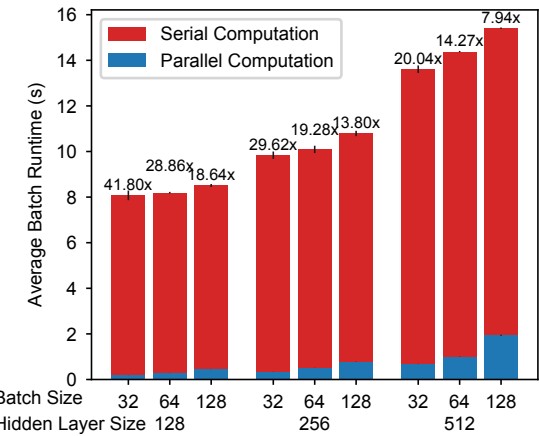

*Figure 10.* Speedup of our parallel SNN implementation over a sequential event-based SNN implementation with exact hard-reset dynamics on the SSC dataset as we vary the hidden dimension and batch size. The model architecture uses 2 feedforward hidden layers and delays. Averages are taken over 100 batches (forward and backward pass) over 3 runs and chunk size is set to 128.

*Table 3.* Average training time (forward and backward pass) per batch in seconds on the SSC dataset for our parallel SNN implementation versus a sequential event-based SNN implementation with exact hard-reset dynamics. The model architecture uses 2 feedforward hidden layers and delays. Averages are taken over 100 batches (forward and backward pass) over 3 runs and chunk size is set to 128.

| Hidden Layer Size | Batch Size | Computation Mode | Avg. Runtime (std. dev.) |
|---|---|---|---|
| 128 | 32 | Parallel | 0.193 (0.003) |
| | | Serial | 8.075 (0.212) |
| | 64 | Parallel | 0.284 (0.003) |
| | | Serial | 8.195 (0.033) |
| | 128 | Parallel | 0.457 (0.004) |
| | | Serial | 8.510 (0.067) |
| 256 | 32 | Parallel | 0.332 (0.004) |
| | | Serial | 9.838 (0.161) |
| | 64 | Parallel | 0.524 (0.004) |
| | | Serial | 10.093 (0.161) |
| | 128 | Parallel | 0.782 (0.006) |
| | | Serial | 10.790 (0.118) |
| 512 | 32 | Parallel | 0.679 (0.004) |
| | | Serial | 13.609 (0.165) |
| | 64 | Parallel | 1.008 (0.007) |
| | | Serial | 14.383 (0.028) |
| | 128 | Parallel | 1.939 (0.023) |
| | | Serial | 15.407 (0.032) |

*Table 4.* Average training time (forward and backward pass) per batch in seconds on the SHD dataset for our parallel SNN implementation versus a sequential event-based SNN implementation with exact hard-reset dynamics. The model architecture uses 2 feedforward hidden layers and delays. Averages are taken over 100 batches (forward and backward pass) over 3 runs and chunk size is set to 128.

| Hidden Layer Size | Batch Size | Computation Mode | Avg. Runtime (std. dev.) |
|---|---|---|---|
| 128 | 32 | Parallel | 0.189 (0.001) |
| | | Serial | 8.352 (0.022) |
| | 64 | Parallel | 0.277 (0.002) |
| | | Serial | 8.167 (0.044) |
| | 128 | Parallel | 0.449 (0.003) |
| | | Serial | 8.614 (0.034) |
| 256 | 32 | Parallel | 0.328 (0.002) |
| | | Serial | 10.087 (0.044) |
| | 64 | Parallel | 0.512 (0.002) |
| | | Serial | 10.316 (0.044) |
| | 128 | Parallel | 0.772 (0.004) |
| | | Serial | 10.514 (0.061) |
| 512 | 32 | Parallel | 0.672 (0.002) |
| | | Serial | 13.828 (0.037) |
| | 64 | Parallel | 1.057 (0.004) |
| | | Serial | 14.530 (0.072) |
| | 128 | Parallel | 1.880 (0.008) |
| | | Serial | 14.955 (0.097) |

*Table 5.* Absolute speed and memory comparison of our method to prior work on the SHD dataset with batch size 256. All methods use 2 hidden layers with delays and 256 or 512 hidden units per layer. * denotes metrics reported in Mészáros et al. (2025) on an RTX A5000; all other numbers were measured on an NVIDIA H100 (80GB). See text for details.

| Method | Epoch Training Time (s) ↓ | Max Memory (MiB) ↓ |
|---|---|---|
| 256 (Hammouamri et al., 2024) | 157.89 | 8,978 |
| 512 (Hammouamri et al., 2024) | 338.94 | 16,366 |
| 256 (Mészáros et al., 2025) | 38.93 | 4,824 |
| 512 (Mészáros et al., 2025) | 141.35 | 8,632 |
| 256 (Ours) | 44.66 | 7,967 |
| 512 (Ours) | 111.55 | 26,184 |

A natural question is how our method compares to optimized prior work in terms of speed and memory. In Table 5 we compare against two discrete-time methods: a surrogate-gradient approach (Hammouamri et al., 2024) and an exact-gradient CUDA implementation (Mészáros et al., 2025). We measured our method and Hammouamri et al. (2024) on an NVIDIA H100 (80GB), while the Mészáros et al. (2025) numbers are taken as reported in their paper on an RTX A5000. Memory for Hammouamri et al. (2024) is PyTorch peak allocated memory (`torch.cuda.max_memory_allocated()`), and ours is peak device memory reported by `jax.profiler`; neither accounts for peak reserved memory. All methods use matched 2-layer delayed models (256 or 512 hidden units), $\Delta t = 1$ms (though ours uses exact spike times—no discretization), and the full SHD input (700 channels). Our method performs favorably in terms of speed compared to Hammouamri et al. (2024), where hardware differences are controlled for, and is competitive in memory. Hammouamri et al. (2024) does not support sub-millisecond spike times and refining the time grid further would substantially increase both compute and memory. In Mészáros et al. (2025), delays are discretized and implemented via per-neuron delay buffers, so holding the maximum delay fixed while decreasing $\Delta t$ increases the number of delay slots (and thus delay-buffer memory) approximately as $1/\Delta t$, and increases the number of time steps that must be computed. We do not claim a hardware-normalized ranking. Table 5 situates our approach in the same practical regime (seconds/epoch, single-digit to tens of GiB) as optimized discrete-time alternatives under matched architectures.

# G. Work Analysis

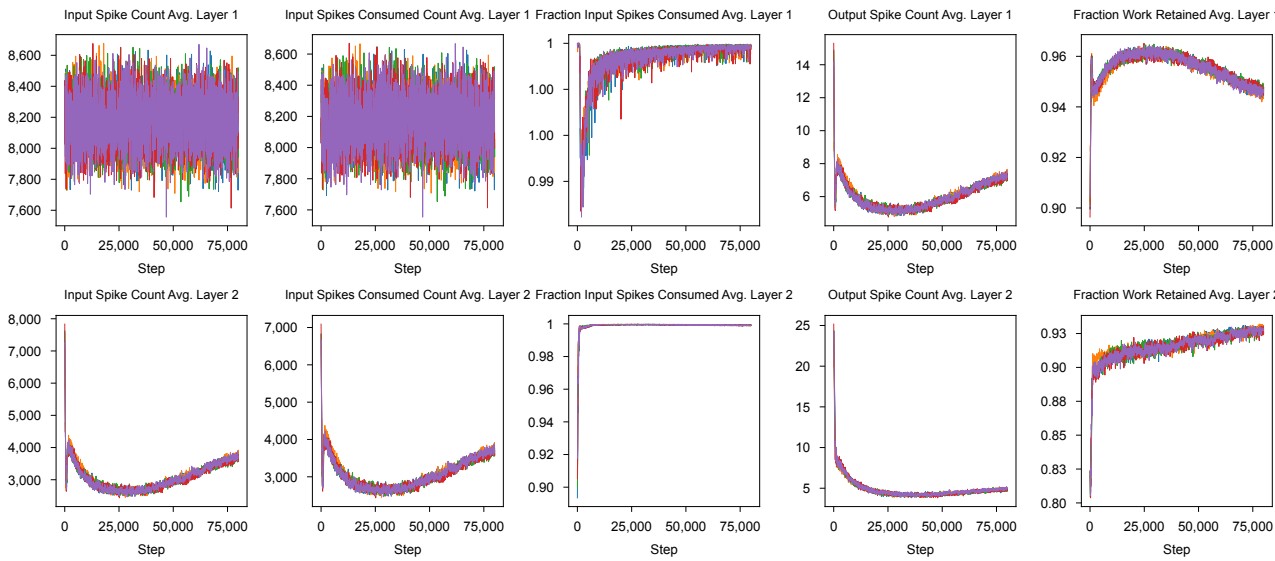

*Figure 11.* Spike metrics for 5 runs on the SSC dataset with 512 hidden units and chunk size 128. On average, layer one and two neurons consume approximately 100% of input spikes. We also note that both layer one and layer two retain a very high percentage of work done (e.g., 80-90%+).

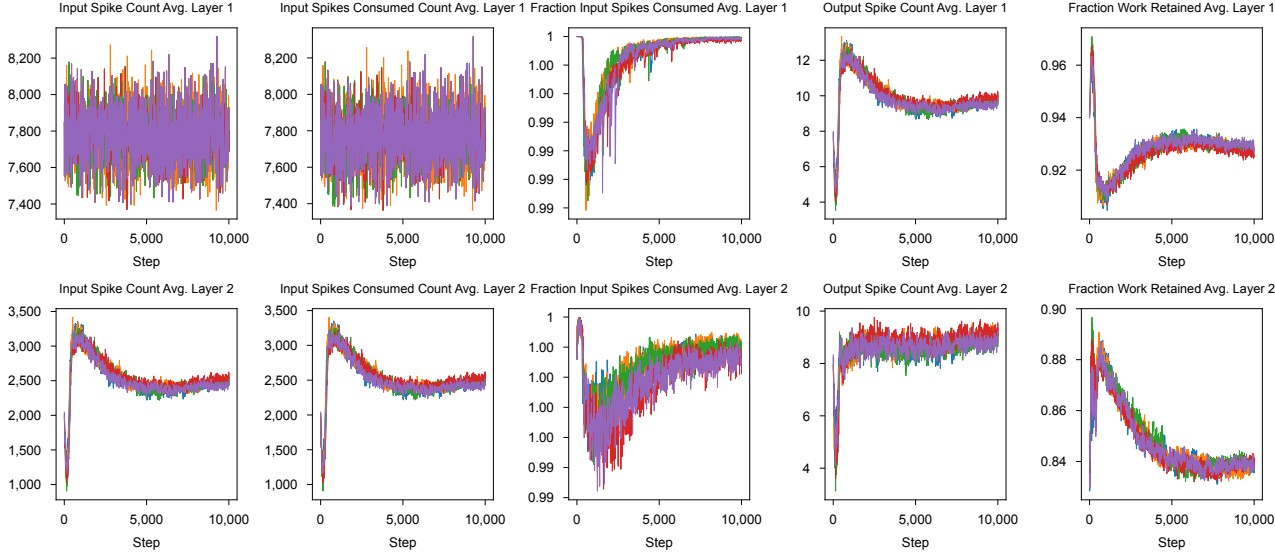

*Figure 12.* Spike metrics for 5 runs on the SHD dataset with 256 hidden units and chunk size 128. On average, layer one and two neurons consume approximately 100% of input spikes. We also note that both layer one and layer two retain a very high percentage of work done (e.g., 80-90%+).

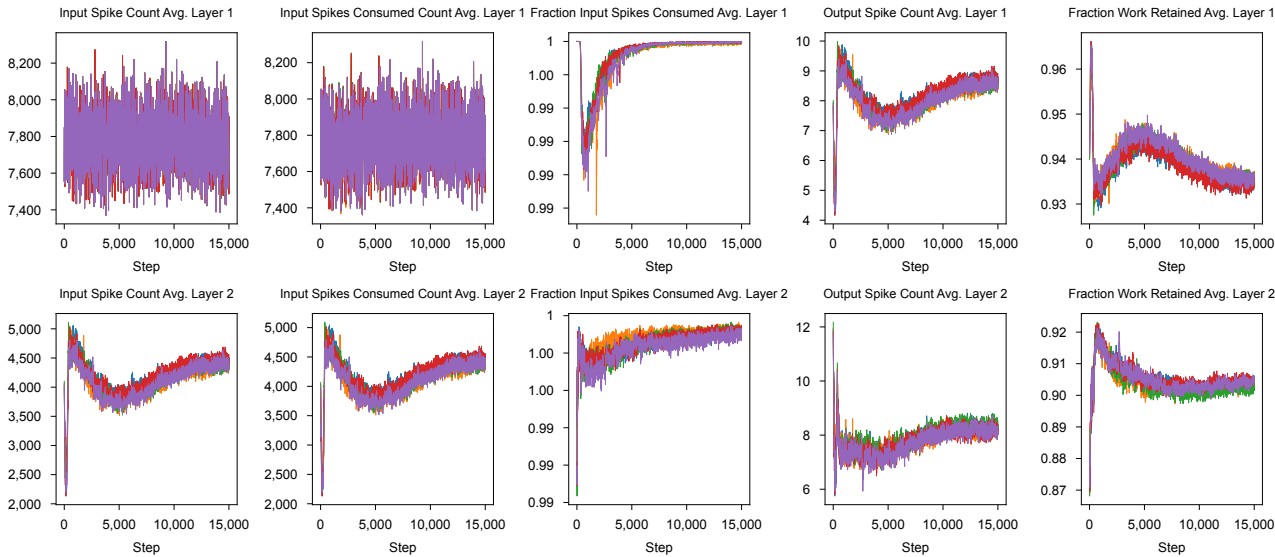

*Figure 13.* Spike metrics for 5 runs on the SHD dataset with 512 hidden units and chunk size 128. On average, layer one and two neurons consume approximately 100% of input spikes. We also note that both layer one and layer two retain a very high percentage of work done (e.g., 80-90%+).

Figures 11, 12, and 13 show key spike metrics for our models. Input Spike Count Avg. is the average number of input spikes received per neuron per batch, Input Spikes Consumed Count Avg. is the average number of input spikes consumed per neuron per batch, Fraction Input Spikes Consumed Avg. is the ratio of consumed spikes to received spikes, Output Spike Count Avg. is the average number of output spikes emitted per neuron per batch, and Fraction Work Retained Avg. is the ratio of the consumed spikes to the number of input spikes processed (see below).

Since our method processes input spikes in chunks, some amount of work is discarded when a neuron spikes early in a chunk. However, we find that this effect is minimal in practice since neurons tend to spike sparsely relative to the number of input spikes they receive. In Figures 11, 12, and 13, we quantify the percentage of work retained during training on the SSC and SHD datasets. In particular, we define the work retention rate as the ratio of the total number of input spikes consumed to the number of input spikes processed. In other words, every time an input spike is visited—which may happen multiple times due to chunking—it contributes one unit of work to the denominator, but only contributes one unit of work to the numerator when it is consumed. So a high ratio implies that spikes are only visited once most of the time, indicating that chunking is not causing excessive wasted work. We find that both layer one and layer two retain a very high percentage of work done (e.g., 80-90%+). The work retained will depend on the chunk size, the number of input spikes, and the firing rates of the neurons. We do not optimize for work retained in this work because our main focus is on accelerating training time.

### G.1. Efficiency Constraints and Regularization

While our chunked parallel processing enables significant speedups in sparse firing regimes, we acknowledge that the theoretical efficiency of this approach is sensitive to the firing rate. In the worst-case scenario—such as a "bursting" regime where a neuron spikes within every chunk—the parallelism reverts to serial execution because the work done in a chunk subsequent to a spike must be discarded and recomputed. However, we argue that this constraint aligns with the fundamental premise of event-based SNNs, which prioritize energy efficiency through sparsity. To ensure our method operates within this optimal regime, we employ a spike-count regularizer (see Appendix E.1) that penalizes excessive activity. This regularizer serves a dual purpose. It promotes biologically plausible sparse representations and explicitly maintains the computational efficiency of the parallel associative scan. In our experiments, this mechanism successfully maintained a high work-retention rate (see Figure 11) confirming that the network learns to utilize the configurations where our parallel method excels, rather than diverging into inefficient high-firing modes. Future work can explore more sophisticated regularization techniques or adaptive chunking strategies to further enhance efficiency across diverse spiking regimes.

