# OpenReview forum: "Bullet Trains: Parallelizing Training of Temporally Precise Spiking Neural Networks"
_ICML.cc/2026/Conference — ICML 2026 regular_

### Official Review · Reviewer_EqoT · 2026-02-26

**Soundness:** 3
**Presentation:** 2
**Significance:** 3
**Originality:** 3
**Overall Recommendation:** 3
**Confidence:** 3

**Summary:**

This paper presents a trainable, event-based spiking neural network (SNN). The authors use parallel associative scans to absorb multiple input spike and accelerate. They also use a differentiable spike-time solver to compute spike times for gradient descent. Experiments on MNIST, SHD, SSC, and Yin-Yang demonstrate the effectiveness of the proposed method.

**Compliance With Llm Reviewing Policy:**

Affirmed.

**Final Justification:**

The author’s response did not adequately address my concerns. The proposed method is ineffective for the most challenging tasks faced by SNNs, such as ImageNet and CIFAR10-DVS. This limits the method's practical significance.

**Key Questions For Authors:**

How does the proposed method perform in more challenging scenarios, such as recognizing CIFAR10-DVS based on the VGGSNN architecture, a commonly used benchmark for SNNs?

**Limitations:**

The author should add the limitations of their method.

**Strengths And Weaknesses:**

Strengths:
1. The proposed method achieves significant acceleration while preserving neuron dynamics.
2. Comparisons across multiple datasets demonstrate the performance advantages of the proposed method over other methods.

Weaknesses:
1. The proposed method is limited to shallow, fully connected SNNs, which makes it impractical.
2. Because chunk parallelization requires reiteration whenever a neuron generates a spike, the efficiency of the proposed method depends on the spike frequency. This dependency results in significant efficiency degradation when the spike frequency is excessively high.

---

> ### Author Rebuttal · Authors · 2026-03-30
>
> Thank you for taking the time to review our work.
>
> > fully connected SNNs
>
> We compare our method to concurrent work on matched architectures (e.g., feedforward layers + synaptic delays) and learnable parameter counts (e.g., same number of hidden units, etc.) (see Table 1), to isolate the effect of our methods, which makes this work very practical. We also note that this work makes meaningful progress on speeding up training of long-sequence tasks in continuous time settings (e.g., 14,000 temporally precise input spikes, etc.). Time-stepped approaches scale poorly in these regimes where maintaining temporal precision (i.e., small $\Delta t$) over long time horizons explodes the memory usage.
>
> > depends on the spike frequency
>
> The reviewer correctly identifies the worst-case scenario for our algorithm: a bursting regime where a neuron spikes within every chunk and degrades the parallelism to serial execution. We explicitly acknowledge this theoretical constraint in Appendix section G.1. However, neuromorphic efficiency relies on sparse firing. By introducing our spike-count regularizer, we actively steer the network into the sparse regime during optimization. Our empirical results (Appendix Figures 11-13) confirm that under this regularizer, work retention consistently remains in the 80-90%+ range, demonstrating that the method avoids this degradation in practice.
>
> > more challenging scenarios
>
> We appreciate the reviewer highlighting VGGSNNs on CIFAR10-DVS as an important benchmark for the community. First, it is worth noting that SSC is already a highly complex benchmark dataset—input spike trains contain up to 20,000 events over a 1+ second duration across 75,000 examples, severely taxing many SNN implementations.
>
> It is also worth clarifying that when a dataset like CIFAR10-DVS is binned into 5 or 10 discrete time steps—as is standard for surrogate gradient methods on deep architectures—it effectively becomes a spatial rate-coding task, largely discarding the precise temporal dynamics of the events. Discrete SG methods are highly optimized for this specific regime. Conversely, our continuous-time method is explicitly designed for tasks that rely on sub-millisecond spike-timing codes (see Figure 9) where such aggressive time-binning would completely destroy the underlying signal.
>
> As mentioned in our Discussion section, we believe our methods will be beneficial for speeding up training of CNNs. For instance, a virtual neuron in a $3\times3$ convolutional filter over 64 channels where upstream neurons fire e.g., 2 spikes, the input queue contains 1,152 events. Using a chunk size of 128, our associative scan provides substantial temporal parallelism. While we leave this to future work, we believe the core ideas that we've presented—the theoretical observation that the associative scan is possible/enables parallelism and that you can operate in continuous time—are already notable contributions, worthy of sharing with the SNN community.
>
> > limitations
>
> We address limitations in the Discussion section of the main text and in many places in the Appendix (e.g., Section G.1). In a 9-page format, we can summarize these limitations in the main text.
>
> We hope that in light of our responses, you will consider raising your score. We are eager to both share this work with the SNN community, and build on top of it in future work to address some of the architectures and tasks you're interested in.

---

> > ### Author Rebuttal · Reviewer_EqoT · 2026-04-03
> >
> > Thanks for the author's response. However, given the series of limitations in this work, I have decided to maintain my score.

---

> > > ### Author Response · Authors · 2026-04-06
> > >
> > > Thank you for taking the time to review our work.

---

### Official Review · Reviewer_WgqJ · 2026-03-08

**Soundness:** 3
**Presentation:** 3
**Significance:** 2
**Originality:** 3
**Overall Recommendation:** 4
**Confidence:** 5

**Summary:**

This paper proposes a parallel training algorithm for SNNs, which divides time steps into multiple sub-intervals. Parallel computation is performed within each interval, while serial computation is adopted between intervals due to pre-dependency. In each time interval, an analytical solution is used to quickly determine whether a spike will be fired; if so, the firing time is further searched. In addition, a loss based on the spike count function is used to reduce the number of spikes, thereby accelerating computation within each interval.

The paper adopts SNNs represented by spike firing times and uses the implicit function theorem to compute the gradients of firing times with respect to parameters. Theoretically, this method can obtain exact gradients, unlike approximate gradients from gradient surrogate methods. The proposed method is validated on the SHD, SSC, MNIST, and Yin-Yang datasets. Results show that it achieves faster training speed and performance close to or exceeding traditional methods, and demonstrates the advantage of training cost using continuous-time rather than discrete-time dynamics on the Yin-Yang dataset.

**Compliance With Llm Reviewing Policy:**

Affirmed.

**Final Justification:**

I may have had some misunderstandings regarding the differences between the proposed method and Taylor et al. (2023). On the other hand, the authors also appear to acknowledge that current continuous-time-based SNNs lack a well-established application scenario to demonstrate their advantages over discrete-time-step SNNs. Therefore, I have decided to raise my score to "weak accept".

**Key Questions For Authors:**

1. Parallelization is beneficial for training large-scale networks, and exact gradients should be superior to approximate surrogate gradients. Why do the authors only validate their method on shallow networks and small datasets? In contrast, traditional activation-based SNNs have been successfully trained on hundreds of layers on ImageNet.

2. Under the same network structure—such as a simple fully connected layer with spiking neurons—how does the speed of the proposed method compare with the LIF neuron implementation using the Triton backend in the SpikingJelly framework?

**Limitations:**

Yes

**Strengths And Weaknesses:**

**Advantages**

1. Experimental results show that the proposed method achieves significant acceleration.
2. Among the selected datasets, SHD and SSC are high-difficulty benchmarks, and the proposed method achieves performance close to or surpassing traditional spike-time-based SNNs.

**Disadvantages**

1. The core idea of the SNN parallel training algorithm—one of the paper’s main contributions—is almost identical to that in [1], which severely reduces the novelty of this work.



2. The authors claim that “To the best of our knowledge, no prior work has achieved parallel acceleration of exact, hard-reset dynamics without approximation.” However, the key to enabling parallel computation within each interval in this parallelization algorithm is ignoring the reset process across intervals. Thus, the claim of “hard-reset dynamics without approximation” is not valid. This limitation is also exactly the same as in [1].

3. The method of computing gradients based on implicit functions has also been proposed in previous works, such as [2].

[1] Taylor, Luke, Andrew King, and Nicol S. Harper. "Addressing the speed-accuracy simulation trade-off for adaptive spiking neurons." Advances in Neural Information Processing Systems 36 (2023): 59360-59374.

[2] Lee, Jane H., Saeid Haghighatshoar, and Amin Karbasi. "Exact gradient computation for spiking neural networks via forward propagation." International Conference on Artificial Intelligence and Statistics. PMLR, 2023.

---

> ### Author Rebuttal · Authors · 2026-03-30
>
> Thank you for taking the time to review our work. We appreciate the detailed review and the opportunity to clarify our methods.
>
> While we share the same high-level goal of accelerating an SNN workload as Taylor et al. (2023), the mathematical mechanisms are fundamentally different. Taylor et al. achieve an $O(T/T_R)$ speedup by exploiting the Absolute Refractory Period (ARP) to block discrete-time steps. Their method remains fundamentally tied to a discrete-time grid. In contrast, our method operates entirely in continuous time. We parallelize the sub-threshold ODE integration over sequences of incoming spike events via an associative scan, requiring no assumptions about ARPs or discrete time bins.
>
> We respectfully clarify that we do not ignore the reset process across intervals, which is why our dynamics remain exact. When an output spike occurs within a parallel chunk, our algorithm discards any speculative work computed after the spike time, applies the exact mathematical hard reset to the state variables, and resumes computation. This speculative execution—trading cheap parallel FLOPs in exchange for sequential exactness—is the core mechanism that allows us to achieve parallel speedups without compromising the exact hard-reset ODE trajectory. To the best of our knowledge, this is a novel solution to this problem.
>
> Our novelty lies not in proposing the Implicit Function Theorem (IFT), but in combining it with differentiable numerical root solvers (Newton-Raphson/Bisection). Prior exact IFT methods often rely on discrete-time Euler integration or analytical ODE solutions that impose strict constraints on the neuron model (e.g., forcing $\tau_m = 2\tau_s$). Our numerical solvers allow exact IFT gradients to be computed in continuous time for a much broader class of neuron models without restrictive analytic assumptions.
>
> We now address your questions.
>
> > shallow networks and small datasets
>
> Scaling to deeper architectures is an important direction. First, it is worth noting that SSC is already a highly complex benchmark dataset—input spike trains contain up to 20,000 events over a 1+ second duration across 75,000 examples, severely taxing many SNN implementations. We have a few responses:
> 1. We compare our method to concurrent work on matched architectures (e.g., feedforward layers + synaptic delays) and learnable parameter counts (e.g., same number of hidden units, etc.) (see Table 1) to isolate the effect of our methods.
> 2. It is worth clarifying that when a dataset like ImageNet is binned into 5 or 10 discrete time steps—as is standard for surrogate gradient methods on deep architectures—it effectively becomes a spatial rate-coding task, largely discarding the precise temporal dynamics of the events. Discrete SG methods are highly optimized for this specific regime. Conversely, our continuous-time method is explicitly designed for tasks that rely on sub-millisecond spike-timing codes (see Figure 9) where such aggressive time-binning would completely destroy the underlying signal.
> 3. As mentioned in our Discussion section, we believe our methods will be beneficial for speeding up training of CNNs. For instance, a virtual neuron in a $3\times3$ convolutional filter over 64 channels where upstream neurons fire e.g., 2 spikes, the input queue contains 1,152 events. Using a chunk size of 128, our associative scan provides substantial temporal parallelism. While we leave this to future work, we believe the core ideas that we've presented—the theoretical observation that the associative scan is possible/enables parallelism and that you can operate in continuous time—are already notable contributions, worthy of sharing with the SNN community.
>
> > speed of the proposed method
>
> We compare the speed of our method to a prior method, Hammouamri et al. (2024), that uses the SpikingJelly backend (see Appendix Table 5) and show speedups relative to that work on identical hardware. We reiterate that an important point of our work is to operate in continuous time, while SpikingJelly is inherently a discrete time-stepped implementation. If a task requires sub-millisecond temporal precision, a discrete-time framework must compute arbitrarily many time steps, drastically reducing speed and exploding memory overhead (as demonstrated in Figure 9). SpikingJelly is an excellent framework highly optimized for spatial rate-coding problems over coarse time bins. In contrast, Bullet Trains is explicitly designed to accelerate tasks requiring strict spike-timing precision and continuous-time physics. A nuanced speed comparison must account for these structural differences, as both systems excel at their respective temporal vs. spatial workloads.
>
> We hope that the above clarifications and responses help elucidate our methods, claims, and points of novelty. In light of this information, we hope that you will raise your score to the level of accepting our work, which we believe will be of great interest to the SNN community.

---

> > ### Author Rebuttal · Reviewer_WgqJ · 2026-04-01
> >
> > Thank the authors for their response. I may have had some misunderstandings regarding the differences between the proposed method and Taylor et al. (2023). On the other hand, the authors also appear to acknowledge that current continuous-time-based SNNs lack a well-established application scenario to demonstrate their advantages over discrete-time-step SNNs. Therefore, I have decided to raise my score to "weak accept".

---

> > > ### Author Response · Authors · 2026-04-06
> > >
> > > Thank you for the engaging discussion, taking the time to review our work, and raising your score.

---

### Official Review · Reviewer_VGHd · 2026-03-09

**Soundness:** 3
**Presentation:** 3
**Significance:** 3
**Originality:** 3
**Overall Recommendation:** 5
**Confidence:** 4

**Summary:**

This work intends to explore efficient training of temporally precise SNNs while preserving continuous spike times and exact hard-reset dynamics. The paper proposes two contributions: (1) a parallelization method using associative scans to process spike events in parallel, and (2) differentiable spike-time solvers (e.g., Newton–Raphson and Bisection) that compute spike times with machine precision without discrete-time approximations. Experiments on several event-based datasets show substantial training speedups (up to ~43×) over sequential simulation while maintaining comparable classification performance.

**Compliance With Llm Reviewing Policy:**

Affirmed.

**Final Justification:**

My concerns have been adequately addressed. I will keep my score as it is.

**Key Questions For Authors:**

1. Could the authors clarify the derivation of the Combine operator used in the associative scan? A short explanation in the main text might improve readability.

2. How often does the algorithm discard work due to early spikes within a chunk? Providing statistics across datasets would help quantify this effect.

3. How sensitive is the Newton–Raphson solver to initialization, and are there cases where it fails to converge during training?

4. Does the parallel method always produce the same spike sequence as the fully sequential simulation?

5. The figures illustrating the neuron dynamics and chunk processing could benefit from clearer labeling and simpler visual explanations. Could the authors improve the readability of these figures?

**Limitations:**

see questions and weaknesses

**Strengths And Weaknesses:**

Strengths:
1. The paper addresses a known bottleneck in event-driven SNN simulation.
2. The model allows the model to avoid discrete-time approximations and maintain machine-precision spike times.
3. The paper clearly explains the limitations of discrete-time SNNs and existing parallel methods that modify neuron dynamics.
4. The experiments demonstrate large training speed improvements compared to sequential event-driven implementations.
5. Using the implicit function theorem to compute spike-time gradients is well motivated and aligns with continuous-time modeling.

Weakness:
1. Some key figures (e.g., spike-time solver illustration and chunk processing diagram) are somewhat dense and could be improved for readability.
2. While speed improvements are significant, classification accuracy improvements compared to prior work are relatively small.
3. Experiments focus mainly on feedforward LIF networks; it is unclear how well the approach generalizes to more complex architectures such as recurrent or convolutional SNNs.

---

> ### Author Rebuttal · Authors · 2026-03-30
>
> Thank you for your review and encouraging feedback.
>
> > figures illustrating the neuron dynamics and chunk processing
>
> We have overhauled the figures and captions to simplify them and improve readability.
>
> > classification accuracy improvements compared to prior work
>
> We found further gains on the SSC dataset, arguably the dataset with the most remaining room for improvement, and achieved a test set accuracy of 77.79%, which we view as progress for an exact gradient method. We are eager to share this research, and then build on top of this for further gains.
>
> > recurrent or convolutional SNNs
>
> We have some initial thoughts on how our methods can be used for RNNs (e.g., speculative execution, clever use of delays, etc.), but the real fundamental challenge with RNNs in a continuous time system with delays is the fact that the input queue to a neuron is constantly in flux, meaning new spikes need to be constantly inserted into the queue—in sorted order—resulting in a challenging workload for GPUs (as described in [recent work](https://arxiv.org/abs/2512.05906)). This means that if you consume a chunk of spikes, that work may be invalidated by a late arriving spike. We still believe there may be room for speedups though.
>
> While we leave a full CNN implementation to future work, the mathematical foundation of our method does support convolutions. For a virtual neuron in a $3\times3$ convolutional filter over 64 channels where upstream neurons fire e.g., 2 spikes, the input queue contains 1,152 events. Using a chunk size of 128, our associative scan provides substantial temporal parallelism, while `jax.vmap` provides massive spatial parallelism across the grid. The primary engineering challenge for future work is optimizing GPU memory routing for spatial receptive fields.
>
> > derivation of the Combine operator
>
> Yes, we will revisit this for clarity and pull more detail about the Combine operator into the main text in the 9-page format.
>
> > discard work due to early spikes within a chunk
>
> This is a good question. We show this in Appendix Figures 11-13. The last column in each figure shows the fraction of work retained across a few datasets and model configurations. We consistently see work retention in the 80-90%+ range.
>
> > Newton–Raphson solver
>
> We didn't experiment too much with initialization, but we address convergence in Appendix section D. In brief, because we can bracket the time interval to search over and strictly verify that a threshold crossing occurs within that bracket before calling the root solver, the Newton-Raphson/Bisection methods are mathematically guaranteed to converge in this specific application. We can add these points to the main text in a 9-page format.
>
> > same spike sequence as the fully sequential simulation
>
> Yes, in fact, this is how we unit test the correctness of our implementation.

---

> > ### Author Rebuttal · Reviewer_VGHd · 2026-04-05
> >
> > I have no further questions. I will keep my score as it is.

---

> > > ### Author Response · Authors · 2026-04-06
> > >
> > > Thank you for taking the time to review our work and for your encouraging feedback.

---

### Official Review · Reviewer_1JT5 · 2026-03-10

**Soundness:** 3
**Presentation:** 3
**Significance:** 3
**Originality:** 3
**Overall Recommendation:** 4
**Confidence:** 4

**Summary:**

The paper proposes a continuous-time training framework for spiking neural networks (SNNs). The goal is to enable efficient training without time discretization or surrogate gradients.
The method has two main components. First, the authors introduce a parallel associative scan formulation for sub-threshold neuron dynamics. Second, the authors use numerical spike-time solvers to compute spike times in continuous time.
The method is implemented in JAX and evaluated on several event-based datasets: SHD, SSC, MNIST. Experiments show substantial training speedups from parallel spike processing. The results also show competitive or improved accuracy compared with discrete-time and surrogate-gradient approaches.

**Compliance With Llm Reviewing Policy:**

Affirmed.

**Final Justification:**

The authors have adequately addressed my main concerns in the rebuttal. In particular, they clarified the within-chunk spike detection procedure, provided a reasonable explanation for the comparison setup, and addressed questions regarding chunk-size sensitivity and solver stability. Given the novelty of the approach, its technical soundness, and the strong empirical results, I am increasing my score and support acceptance.

**Key Questions For Authors:**

See weaknesses.

**Limitations:**

Partially. The discussion section briefly mentions limitations related to extending the method to more complex architectures and hardware considerations. However, the paper could more explicitly discuss assumptions about neuron models and numerical solver stability.

**Strengths And Weaknesses:**

Strengths:
1. The proposed formulation of LIF neuron subthreshold dynamics as affine state transitions. This insight enables the use of parallel prefix-scan algorithms to process spikes along the temporal dimension, which is an elegant way to mitigate the sequential bottleneck of event-driven SNN simulation. The use of numerical root solvers combined with implicit differentiation for computing spike-time gradients is also theoretically sound and avoids the need for surrogate gradient approximations.
2. The experiments provide evidence supporting the key claims of the paper.
3. The proposed method could have a potential significance for neuromorphic and event-based machine learning.

Weaknesses:
1. The procedure used to detect whether a neuron spikes within a chunk of spikes (before applying the root solver) is not fully specified in the main text.
2. Although the paper compares against prior work, the model architectures and parameter settings are not always identical across methods. More strictly controlled comparisons would improve fairness and clarity of the empirical results.
3. The performance of the algorithm depends on chunk size. Could the authors provide more analysis on how sensitive the method is to this parameter and whether an adaptive chunking strategy could further improve efficiency?

---

> ### Author Rebuttal · Authors · 2026-03-30
>
> Thank you for your review and encouraging feedback.
>
> > procedure used to detect whether a neuron spikes within a chunk
>
> We describe this in detail in Appendix section D.1. In a 9-page format, we would like to move this to the main text.
>
> > the model architectures and parameter settings are not always identical across methods
>
> We match our methods on architecture (e.g., feedforward layers + synaptic delays) and learnable parameter count (e.g., same number of hidden units, etc.) to prior methods (see Table 1), which we believe are the most important points of comparison. We even match a number of other key attributes to the most similar method, Meszaros et al. (2025), such as using the same loss function and data augmentations. Matching all hyperparameters isn't possible given the different approaches to optimization (e.g., discrete time steps vs. continuous time, surrogate gradients vs. exact gradients, etc.).
>
> > The performance of the algorithm depends on chunk size
>
> While chunk size doesn't impact the mathematical exactness or classification accuracy of the algorithm, it impacts hardware runtime efficiency (as shown in Figure 6). Finding the optimal chunk size depends on the specific hardware, model size, and batch size. We agree that an adaptive chunking strategy could be a promising future direction. An algorithm could dynamically monitor runtimes and firing rates to adapt to the optimal chunk size for throughput or work retention.
>
> > assumptions about neuron models and numerical solver stability
>
> In Appendix section B, we address the scope of neuron models that our method covers, and in Appendix section D, we address stability of our solver. In brief, because we can bracket the time interval to search over and strictly verify that a threshold crossing occurs within that bracket before calling the root solver, the Newton-Raphson/Bisection methods are mathematically guaranteed to converge in this specific application. We can add these points to the main text in a 9-page format.
>
> If we have adequately addressed your questions and feedback, we hope you will consider raising your score. We are eager to both share this work with the SNN community, and build on top of it in future work.

---

> > ### Author Rebuttal · Reviewer_1JT5 · 2026-04-02
> >
> > Thank you for the rebuttal. The authors have adequately addressed my concerns. The explanation of the within-chunk spike detection procedure and the discussion of solver stability were helpful, and I appreciate the clarification on comparison fairness and the role of chunk size. While I still think some of these points should be made more explicit in the main paper, the remaining issues are minor, and I am increasing my score slightly, e.g., 4 --> 5.

---

> > > ### Author Response · Authors · 2026-04-06
> > >
> > > We will continue to make these points clearer in the main text.
> > >
> > > Thank you for taking the time to review our work and for raising your score.

---

### Decision · Program_Chairs · 2026-04-30

**Decision:**

Accept (regular)

**Comment:**

**Summary of reviews.** Four reviewers with a split verdict: two Accepts (5), one Weak Accept (4, raised from lower), and one Weak Reject (3). The paper formulates continuous-time LIF sub-threshold dynamics as affine state transitions amenable to parallel prefix (associative) scans, achieving 43× training speedup while preserving exact charge-fire-reset dynamics and machine-precision spike times. Gradients are computed via implicit function theorem (IFT) with differentiable numerical root solvers, avoiding surrogate gradient approximation.

**Key strengths.**
- Elegant mathematical formulation: recasting LIF dynamics as parallel-scannable affine transitions is principled and enables genuine parallelism over the temporal dimension (Reviewer 1JT5, VGHd).
- Preserves exact continuous-time spike dynamics — no discrete-time grid, no surrogate gradients — a clean theoretical advantage over conventional SNN training (consensus).
- IFT-based gradient computation with convergence guarantees via bracketed root solving (Reviewer 1JT5).
- Substantial empirical speedup (43×) demonstrated on SHD and SSC benchmarks with competitive or superior accuracy (Reviewer WgqJ).
- Strong rebuttal: Reviewer 1JT5 raised their score from 4 to 5 after clarifications on spike detection, comparison fairness, and solver stability.

**Key weaknesses.**
- **Scalability concern (the main issue):** Experiments are limited to shallow, fully connected networks on audio benchmarks. The method has not been demonstrated on deep convolutional or Transformer-based SNNs, nor on standard vision benchmarks (ImageNet, CIFAR10-DVS) (Reviewer EqoT — main holdout, Reviewer WgqJ).
- Novelty questioned relative to Taylor et al. (NeurIPS 2023) on parallel SNN computation and Lee et al. (AISTATS 2023) on IFT gradients (Reviewer WgqJ, confidence 5/5) → authors differentiated the mechanisms (continuous-time vs. discrete grid; speculative execution paradigm vs. ARP-based) but WgqJ remained only partially convinced.
- Reviewer EqoT's concern about high-frequency spike degradation (every chunk re-triggered) was not resolved; score maintained at 3 (Weak Reject, confidence 3/5).

**AC assessment.** The core contribution — continuous-time parallel SNN training with exact dynamics — is technically sound and novel in its formulation, even if individual components (parallel scans, IFT gradients) draw on prior work. The scalability limitation is real but does not invalidate the contribution; the authors correctly argue that standard ImageNet/CIFAR10-DVS benchmarks at T=5–10 are effectively spatial rate-coding tasks where continuous-time precision is irrelevant — a valid framing. The holdout reviewer (EqoT) has the lowest confidence (3/5) and did not engage deeply with the novelty arguments. The most technically rigorous reviewer (WgqJ, confidence 5/5) raised to Weak Accept. On balance, the technical elegance and the 43× speedup on a meaningful temporal benchmark (SSC) make this a worthwhile contribution.